



# Sensitivity of land-atmosphere coupling strength to perturbations of early-morning temperature and moisture profiles in the European summer

Lisa Jach, Thomas Schwitalla, Oliver Branch, Kirsten Warrach-Sagi, Volker Wulfmeyer

Institute of Physics and Meteorology, University of Hohenheim, Stuttgart, Germany

*Correspondence to*: Lisa Jach (lisa.jach@uni-hohenheim.de)

**Abstract.** Land-atmosphere coupling can have a crucial impact on convective initiation. Yet, uncertainty remains in the analyses of the atmospheric segment of the coupling between land surface wetness and the triggering of deep moist convection,

particularly over Europe. One reason for this is a lack of suitable data. To overcome this limitation, we perturb early-morning temperature and moisture profiles from a regional climate simulation covering the period 1986-2015 over Europe to create a spread in atmospheric conditions. Applying the 'Convective Triggering Potential – low-level Humidity Index' framework, we analyze whether and how strongly the coupling strength and the predominance of positive versus negative feedbacks are sensitive to modifications in the atmospheric conditions.

The results show that the hotspot region in northeastern Europe, in which strong feedbacks are likely to occur, is insensitive to temperature and moisture changes, but the number of potential feedback days varies by up to 20 days per season in dependence of the atmospheric background conditions. Temperature modifications rather control differences in the coupling strength in the north of the domain, while moisture changes dominant the control in the south. In the north of the hotspot region, a predominance for positive feedbacks (deep convection over wet soils) remains, but a switch of the dominant feedback

class between positive feedbacks and a transition zone (convection over any soil, but usually shallow convection) occurred from the Alps to around the Black Sea. This indicates a dependence of the dominant feedback class on temperature and relative humidity in this region.

## 1    Introduction

Land-atmosphere (L-A) coupling plays a key role for understanding states in the climate system. L-A coupling means the

covariance in the land and atmospheric states. It shapes e.g. the atmospheric water and energy cycles, and through this influences the intensity and duration of extreme events such as heat waves (Ukkola et al., 2018; Jaeger and Seneviratne, 2011; van Heerwaarden and Teuling, 2014), drought periods (Miralles et al., 2019), or the occurrence of heavy rainfall events. Furthermore, the feedback processes influence the climate response to land surface modifications (Hirsch et al., 2014; Laguë et al., 2019) suggesting importance of the processes' accurate representation in climate models to improve projections.



The local coupling (LoCo) process chain outlines the connection between soil moisture and precipitation through the turbulent surface fluxes modifying the evolution of the boundary layer (BL), and finally, leading to different conditions for cloud and precipitation formation (Santanello et al., 2009, 2011). Various coupling metrics have been developed to investigate the nature and intensity of this and other relationships in the climate system (Santanello et al., 2018). Individual processes in the chain exhibit different intensities and the feedback sign can diverge in dependence of the region (e.g. Findell et al., 2011; Findell &

Eltahir, 2003; Knist et al., 2017; Koster et al., 2004) and the period of time investigated. Coupling hotspots mainly occur in transition regions between dry and wet climates (e.g. Gentine et al., 2013; Koster et al., 2004; Taylor et al., 2012). Temporal variability is apparent at interannual scales (Guo and Dirmeyer, 2013; Lorenz et al., 2015) and in trends of the coupling strength (Dirmeyer et al., 2013, 2012; Seneviratne et al., 2006).

Uncertainty remains in the accurate quantification of the coupling strength along the LoCo process chain, especially in the

atmospheric segment. From the physical perspective, the strength is influenced by both the prevailing land surface and the atmospheric state. Jach et al. (2020) showed that extreme afforestation led to weaker coupling between surface moisture and convection triggering, and a less pronounced favor for convection triggering over wet soils in the European summer. The conversion of current vegetation to grassland had the opposite effect. However, Davin et al. (2020) showed that the same land use and land cover change scenarios as used in Jach et al. (2020) initiated different responses in near-surface temperature

within the ensemble of regional climate models from the Flagship Pilot Study 'Land-Use and Climate Across Scales' (LUCAS) owing to deficiencies in the computation of evapotranspiration. Understanding potential implications of these uncertainties for impacts of land use and land cover changes on L-A coupling strength and climate variability was one motivation of our study. From the technical perspective, the coupling strength is influenced by the choice of the data set used for the investigation (Dirmeyer et al., 2018; Ferguson and Wood, 2011) and, in case of models, their configuration such as parameterization schemes

(Chen et al., 2017; Milovac et al., 2016; Pitman et al., 2009), initialization (Santanello Jr. et al., 2019) or model resolution (Hohenegger et al., 2009; Knist et al., 2020; J. Sun & Pritchard, 2018; Jian Sun & Pritchard, 2016; Taylor et al., 2013). Studies on the regional scale over Europe often use a single model (Baur et al., 2018; Jach et al., 2020; Lorenz et al., 2012) or target only the terrestrial segment (soil moisture-surface flux coupling) of the local coupling process chain (Knist et al., 2017). Coordinated model intercomparison studies such as the Global Land-Atmosphere Coupling Experiment (GLACE)-Initiative

apply general circulation or earth system models (Guo et al., 2006; Koster et al., 2006, 2011; Comer and Best, 2012). On the one hand, this circumvents the need to use lateral BL. On the other hand, the horizontal resolution of these model runs is usually in the order of 1° to 2° grid spacing. This reduces the models' ability to represent detailed surface characteristics. These, in turn, play a key role for triggering convection e.g. due to differential heating.

The 'Convective Triggering Potential – Low-Level Humidity Index' (CTP-HI$_{low}$) framework (Findell and Eltahir, 2003a, b) is

a commonly used process-based coupling metric to investigate the link between surface moisture and convection triggering. It bases on the hypothesis that the structure of the early-morning BL (atmospheric pre-conditioning) gives an indication about the likelihood for locally triggered afternoon precipitation over differently wet soils. Later works added soil moisture (Roundy et al., 2013) or the evaporative fraction (Findell et al., 2011; Berg et al., 2013) as a third dimension. Efforts have been made



to test the global applicability of the framework, which make use of climatologies of the metrics (Ferguson and Wood, 2011;
Wakefield et al., 2019).

Analyzing the atmospheric segment on a process-based level requires information about the vertical structure of the atmosphere. The data requirements for studying the atmospheric segment of L-A coupling on the process-level and in a spatially explicit way can be summarized as follows: vertical temperature and moisture profiles are needed 1) with a sufficiently long data record (period of at least 12 summers for metrics targeting convection triggering), to comply to the data
length requirements for robust results (Findell et al., 2015), 2) with a high enough temporal resolution to be able to extract the time step close to the local sunrise, and 3) increasing vertical resolutions improve the estimate (Wakefield et al., 2021). These high requirements limit the datasets available for a study on the continental scale for Europe. Observations of early-morning vertical temperature and moisture profiles are rare and usually point measurements. The typical radiosonde launch times (00 UTC and 12 UTC) do not cover the early-morning hours over Europe. Other observational products such satellite-based profil
data often have coarse vertical resolutions (Wulfmeyer et al., 2015). Thus, the reliance on model data. However, the lack of suitable observations challenges the validation of results, which provides the incentive for building up a network of coordinated measurement sites like the Land-Atmosphere Feedback Observatory (LAFO) of the University of Hohenheim (Wulfmeyer et al., 2020; Späth et al., 2019).

To study the sensitivity within the atmospheric segment of L-A coupling strength to differences in the atmospheric pre-
conditioning, we developed a perturbation approach in which temperature and moisture were systematically increased or decreased. The approach is based on the hypothesis that the temperature and moisture fields can diverge in their mean, as well as their vertical, temporal, and horizontal distributions, and the framework only recognizes the differences regardless of their origin. Hence, we are able to approximate a range in coupling strength of the atmospheric segment by systematically modifying the temperature and moisture fields before applying the CTP-HI$_{low}$ framework. Here, we focus on differences in the mean and
the vertical distribution. Temperature modifications at the surface range between ±2 K which is derived from an acceptable range of near-surface temperature biases occurring in climate simulations as defined by Kotlarski et al. (2014). The modifications decrease over height. The perturbations of moisture were implemented under consideration of the close relationship between temperature and water vapor in the atmosphere, thus, taking into account the respective temperature perturbation (e.g. Willett et al., 2010; Bastin et al., 2019).

With this approach we focus on two research questions:

1) How sensitive is the L-A coupling strength to modification of temperature and moisture profiles during the European summer months (JJA)?

2) Where can we identify reliable L-A coupling hotspots over Europe?

The paper is structured as follows: Section 2 describes the data set and analysis methods applied. This is followed by the
analysis of the impacts of temperature and moisture modifications on estimates of L-A coupling strength over Europe in section 3. The discussion of the results follows in section 4, and finally, in section 5 we summarize our findings, potential implications and provide an outlook on future research.



## 2 Materials and Methods

### 2.1 Data

#### 2.1.1 Model data

The data base for the following analysis is a model simulation of Jach et al. (2020) hereafter named CTRL. It is a regional climate simulation on a 0.44° grid increment conducted with the Weather Research and Forecasting (WRF) model Version 3.8.1 (Skamarock et al., 2008; Powers et al., 2017) coupled to the NOAH-MP land surface model (Niu et al., 2011). The applied parameterizations are summarized in Tab. 1. The simulation was forced with ERA-Interim reanalysis data from the
European Centre for Medium-Range Weather Forecasts (ECMWF) (Dee et al., 2011) for the period 1986-2015 over the EURO-CORDEX domain (Jacob et al., 2020). The vegetation map is based on the CORINE land cover classification from 2006 (European Environmental Agency, 2013), and the soil texture was derived from the Harmonized World Soil Database at 30 arcsec grid spacing (Milovac et al., 2014). The simulation is part of the model ensemble of the regional model-intercomparison project LUCAS. LUCAS investigates impacts of the implementation of land use and land cover changes in
regional climate simulations.

Table 1: Applied parameterizations of the simulations from Jach et al. (2020).

| Model physics | Parameterization scheme |
|---|---|
| Microphysics Scheme | New Thompson scheme (Thompson et al., 2004) |
| Short-Wave Radiation Scheme | Rapid Radiative Transfer Model (RRTMG) scheme (Iacono et al., |
| Long-Wave Radiation Scheme | Rapid Radiative Transfer Model (RRTMG) scheme (Iacono et al., 2008; Mlawer et al., 1997) |
| Boundary Layer Scheme | MYNN Level 2.5 PBL (Nakanishi and Niino, 2009) |
| Convection Scheme | Kain-Fritsch scheme (Kain, 2004) |
| Land Surface Model | NOAH-MP land surface model (Niu et al., 2011) |
| Surface Layer Scheme | MYNN surface layer scheme (Nakanishi and Niino, 2009) |

### 2.2 CTP-HI$_{low}$ framework

The coupling metric "Convective Triggering Potential" – "Low-level humidity index" (CTP-HI$_{low}$) framework (Findell and Eltahir, 2003a, b) was used to estimate the coupling strength between land surface moisture and convection triggering. It utilizes vertical temperature and moisture profiles around sunrise to calculate an atmospheric stability (CTP) and humidity deficit (HI$_{low}$) measure.





CTP depicts the divergence of the temperature profile from the moist adiabatic lapse rate integrated between 100 hPa to 300
hPa above ground level (AGL) and is given in the unit [J kg$^{-1}$]. Its calculation is analogous to that of CAPE for the predefined
layer using modeled air temperature. Analyzing this specific layer follows the hypothesis that the BL top is almost always
incorporated, and hence, differences in the atmospheric structure may reveal differences in the likelihood for convection
triggering. The pressure height estimates are valid for Europe, but maybe limit investigations of the pre-conditioning in hot
and arid regions, where the BL usually grows to higher altitudes through-out the day. However, the variables CTP and HI$_{low}$
have been used in combination with wind shear before within arid regions with good predictive skill for convection initiation
triggered by differential surface heating (e.g. Branch and Wulfmeyer, 2019). Large CTP values denote strong divergence of
the temperature profiles from the moist adiabat, and hence greater instability. Small but positive values indicate temperature
profiles that are close to the moist adiabat, i.e. conditionally unstable, and negative CTP values indicate a temperature inversion
in the layer between 100 to 300hPa above ground, which would inhibit deep convection and the formation of precipitation
throughout the subsequent day.

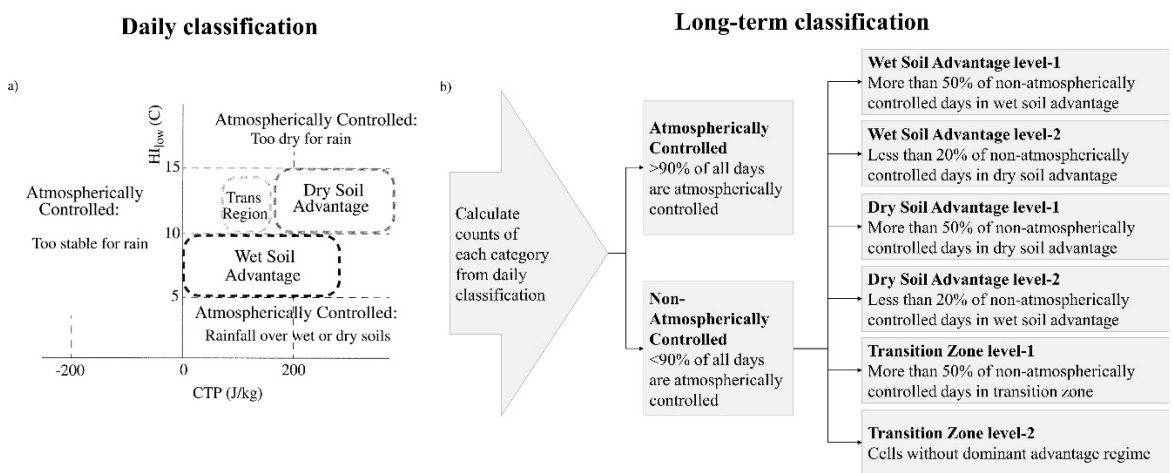

**Figure 1: Schematic depicting the coupling strength classification with the convective triggering potential - low-level humidity (CTP-HI$_{low}$) framework by Findell and Eltahir (2003a,b) (adopted from Jach et al. 2020). a) shows the threshold values from Findell and Eltahir (2003a); their Fig. 15. b) summarizes the approach for the long-term classification as explained in Findell and Eltahir (2003b).**

The HI$_{low}$ measures the dew point depression at 50 hPa and 150 hPa AGL and has the unit [°C]:

$$HI_{low} = \left( T_{p_{sfc}-50hPa} - T_{d,p_{sfc}-50hPa} \right) + \left( T_{p_{sfc}-150hPa} - T_{d,p_{sfc}-150hPa} \right), \tag{1}$$

Where $T_{p_{sfc}-50hPa}$ is the temperature at 50hPa AGL and $T_{d,p_{sfc}-50hPa}$ the dew point temperature at 50hPa AGL. Equivalently,

$T_{p_{sfc}-150hPa}$ and $T_{d,p_{sfc}-150hPa}$ are the temperature and dew point temperature, respectively, at 150hPa AGL.



CTP and $HI_{low}$ form the basis for categorizing early-morning BL conditions on a daily basis in (1) prone for triggering convection over wet or (2) dry soils, (3) a transition zone between wet and dry advantage, or (4) conditions inhibiting a contribution of the land surface to the triggering of deep convection. In the latter case, the occurrence of precipitation is purely atmospherically controlled (AC). This can have three causes: Either the BL is very humid ($HI_{low} < 5°C$) and rainfall is just as likely to occur over any surface, or the BL is very dry ($HI_{low} > 15°C$) and moist convection and precipitation rarely occur in general. Finally, when the BL is stable ($CTP < 0 J\ kg^{-1}$), deep convection is inhibited by an inversion, only shallow clouds can occur. The first three defined categories (1)-(3) are jointly considered as non-atmospherically controlled (nAC). These indicate the percentage of days within the study period with high potential for feedbacks of any kind. Triggering convection over wet soils (1) follows the hydrological pathway meaning positive soil moisture-evapotranspiration-precipitation feedbacks. Hence, greater soil moisture leads to a moistening of the BL through evapotranspiration, and more precipitation. Conversely, triggering convection over dry soils (2) occurs along the thermal triggering pathway during which a high sensible heat flux leads to boundary layer growth and upward mixing of moist air to heights where condensation and formation of rainfall can occur (Dirmeyer et al., 2014). In the transition zone, convection can be triggered over wet or dry soils, though, no convection is the most likely outcome. Here we apply the original threshold values from Findell and Eltahir (2003), which are shown in Fig. 1a. The daily categorizations are then used to derive a long-term coupling regime for each grid cell, based on the relative occurrence of each category in the study period (Fig. 1b). At first, a cell with more than 90% of the day during the study period under atmospheric control is defined as AC. If this is not the case, the partitioning of the nAC days in wet and dry soil advantage, as well as transition zone days is used to determine the dominant feedback class. A level-1 feedback regime denotes that >50% of the nAC days in the cell are in the respective feedback class. Level-2 wet or dry soil advantage means that less than 20% of the respective other class occurs in the cell during the study period, while level-2 transition zone covers all cells remaining unlabeled.

## 2.3 Perturbation approach

Early-morning profiles of temperature and moisture are required to compute the CTP-$HI_{low}$ framework investigating the pre-conditioning for convection triggering during the day. Due to the large expansion of the domain covering several time zones, at the same time (UTC) the BL evolution on the eastern edge of the domain is in a different stage as that of the western edge which can lead to substantial differences in the results of the coupling metric (Wakefield et al., 2021). Hence, the accurate UTC time step to depict the pre-convective BL for the coupling assessment cannot be unified throughout the domain. To ensure this comparability between eastern and western Europe, we determined the sunrise hour in the model using shortwave downward radiation. The profiles were extracted for the UTC time step in which shortwave downward radiation exceeded a value of zero the first time for each day and cell. The profiles from model output around local time sunrise of each day serve as basis for the perturbation analysis. In the following section, we describe how the profiles were perturbed. The approach is based on the hypothesis that the temperature and moisture fields can vary in terms of their mean, and their horizontal, vertical



and temporal distributions. During the perturbations, we investigate the impact of modifying the mean and the vertical distribution. The temporal and horizontal distributions were not modified, although e.g. warming is known to widen and flatten the distribution of temperature over time, and therefore, slightly change the shape of the distribution. The processes and mechanisms leading to a change in the temporal distribution are complex and non-linear, meaning that they cannot be reproduced easily by perturbations. Differences in the spatial distribution (such as warmer conditions in France with colder conditions over Eastern Europe) were not specifically depicted. The CTP-HI$_{low}$ framework utilizes single columns and does not recognize horizontal connections.

### 2.3.1 Temperature perturbations

The temperature profiles were modified by adding a constant temperature (T)-factor in Kelvin to the daily profiles. The factor is fixed in time, homogeneous over the domain, and decreases with altitude. Decreasing the impact over height follows the hypothesis that a surface temperature change does not propagate evenly throughout the atmospheric column. The T-factor for each atmospheric layer was derived using a simple linear regression model and calculating the mean coefficient of determination for each atmospheric layer. Therefore, it corresponds to the fraction of variance in temperature for each atmospheric layer explainable by the temperature variance at the surface.

The first set of temperature perturbations (hereafter called core set) captures differences in the mean air temperature near the surface and in the vertical by applying the temperature factor. In this case, the perturbation amounts to ±2 K at the surface (= T-factor*2). This range was derived from the acceptable range of biases in temperatures in Kotlarski et al. (2014). A second set of perturbations served to investigate the effect of differences in the shape of the profiles (e.g. greater or smaller inversions) leading to divergence of the gradients. For this purpose, we determined the divergence of the mean temperature profiles of summers with the highest near-surface temperature or near-surface moisture anomalies from the mean temperature profile of all thirty years to produce five divergence T-factors. Chosen were the summers with (1) minimum (cold) and (2) maximum (hot) near-surface temperature, as well as the summers with the (3) minimum (dry) and (4) maximum (wet) near-surface relative humidity, as well as (5) maximum near-surface specific humidity (wet_abs). The year with the minimum near-surface specific humidity corresponds to the cold summer. Table 2 summarizes the years chosen for the divergence T-factors and the sign of their temperature and moisture anomalies, respectively. These were added to the temperature profiles from the CTRL run on a daily basis. In a second step, the divergence-cases were further modified by adding the same factor used for the core set in order to investigate the effect of differences in the gradient with additional surface warming or cooling on the coupling strength. Larger perturbation factors up to ±5K were tested, which led to similar patterns of differences, and diverged only in the magnitude of the impact (not shown).



**Table 2: Anomalies from the JJA mean of the CTRL run in temperature and moisture in years chosen as basis for the alternative factors; * the cold and dry_abs are the same year.**

|  | Negative T-anomaly | | Positive T-anomaly | | |
| --- | --- | --- | --- | --- | --- |
| **Negative q-anomaly** | cold/dry_abs* (1986) | dry (1994) | -- | | |
| **Positive q-anomaly** | -- | | hot (2003) | wet_abs (2010) | wet (2013) |

### 2.3.2 Moisture perturbations

Besides the temperature, also the moisture content in the atmosphere is expected to have an impact on the coupling strength.
Willett et al. (2010) investigate the scaling of concurrent temperature and moisture changes for different regions around the globe based on observations and models. For the northern hemisphere, they found that temperature and moisture are strongly positively correlated and that 1 K temperature changes corresponds to on average 8.81% change in moisture. The factor for northern (9.66% $K^{-1}$) and southern (7.74% $K^{-1}$) Europe slightly deviate. Under the assumption that the scaling is valid through the entire atmospheric column, the northern hemisphere factor was used for the moisture perturbations. Hence, the magnitude
is dependent of the respective temperature perturbation and the moisture present in the atmosphere in the CTRL. This ensures two things: First, the relation of temperature and moisture is maintained, and second, the higher atmospheric layers do not experience unrealistic increases in moisture, which could have occurred using fixed factors. As for the temperature perturbations, the mean moisture and the shape of the profiles were modified, but the temporal and spatial variance were not. To further prevent the development of unrealistically high moisture content in the atmosphere in humid regions, the saturation
vapor pressure was determined for the temperature after perturbation and used to cap the moisture increase. Negative moisture content was prevented by setting a lower boundary of 0 g $kg^{-1}$. Thus, the relative humidity (in terms of specific humidity/sat. specific humidity) is designed to remain between 0 and 100% in all atmospheric layers.

### 2.4 Statistical sensitivity assessment

A sensitivity index was used to achieve a grid wise estimate whether temperature modifications or moisture modifications
have a higher impact on the corresponding variable. The index compares the magnitude of differences in a variable x caused by modifying moisture or temperature only from the CTRL. The approach is described using the following formula:

$$x_{sens} = \frac{\Sigma\left(\left(x_{Q_{low}}-x_{ref}\right)^2+\left(x_{Q_{hi}}-x_{ref}\right)^2\right)-\Sigma\left(\left(x_{T_{low}}-x_{ref}\right)^2+\left(x_{T_{hi}}-x_{ref}\right)^2\right)}{\Sigma\left(\left(x_{Q_{low}}-x_{ref}\right)^2+\left(x_{Q_{hi}}-x_{ref}\right)^2\right)+\Sigma\left(\left(x_{T_{low}}-x_{ref}\right)^2+\left(x_{T_{hi}}-x_{ref}\right)^2\right)} \quad (2)$$

with $x_{ref}$ representing the value of the unperturbed case, $x_{Q_{low}}$ is the value of the perturbation case of isolated decrease in moisture, $x_{Q_{hi}}$ is the case with an isolated increase in moisture, $x_{T_{low}}$ is the case with an isolated decrease in temperature, and
$x_{T_{hi}}$ is the case with an isolated increase in temperature, respectively. Thus, the perturbation cases with isolated temperature or



moisture modifications were used for this analysis. The index was then normalized to a value between -1 and 1 by dividing the squared sum of differences induced by moisture changes minus the squared sum of differences induced by temperature changes by the total squared sum of differences from the CTRL in all cases. A sensitivity index close to -1 indicates a strong temperature control on the variable, while a sensitivity index close to 1 indicates a strong moisture control. With a sensitivity

index around 0, moisture and temperature variations have an equal impact on changes in x.

In this study, we used the temperature perturbation of ±2 K, and the cases with the corresponding moisture perturbations of ±2*8.81% K$^{-1}$, from the core-perturbations set to estimate the relative importance of temperature versus moisture changes for CTP, HI$_{low}$, and the occurrence of nAC-days, wet and dry soil advantage as well as transition zone days. We limited the analysis to regions, where on average at least 2 days per summer (~2.5% of the summer days) are in the respective category.

## 2.5    Uncertainty of hotspot location and feedback sign

Two measures were used to depict the sensitivity of the long-term feedback regimes in the perturbation cases. The first metric $I_{feed}$ measures the degree of agreement of the long-term classification based on the CTP-HI$_{low}$ framework among the perturbation cases with that of the CTRL case. A value close to 1 indicates that nearly all perturbations had the same long-term classification no matter which perturbation factors were applied. A value close to 0 indicates an overall disagreement in

the long-term classifications with the CTRL case. The classification is sensitive to differences in the temperature and moisture profiles.

$$I_{feed} = 1 - \frac{\sum_1^n (cat_n \neq cat_{CTRL})}{n}, \tag{3}$$

With $\sum_1^n (cat_n \neq cat_{CTRL})$ denoting the sum of perturbation cases in which the long-term classification disagrees with that of the CTRL case, and n being the number of all perturbation cases tested. A second metric $I_{cat}$ was used to quantify the share

of perturbation cases in which each of the feedback categories occurred. It was determined for nAC-days, and days in wet soil advantage, dry soil advantage or transition zone. Level-1 and level-2 cells of the feedback categories were grouped together before deriving the metric.

$$I_{cat} = \frac{\sum n_{cat}}{n}, \tag{4}$$

With $n_{cat}$ being the number of perturbation cases in the respective category. A value of 0 denotes that the category was never

dominant and a value of 1 denotes that the category was always dominant.

## 3    Results

### 3.1    Perturbation analysis

In this chapter we describe how differences in the mean temperature and moisture profiles impact the frequency of favorable conditions for local land surface triggered deep convection, how the likelihood for convection triggered over wet versus dry





soils changes, and how these influences are represented in classifications of long-term feedback regimes with the CTP-HI$_{low}$ framework.

### 3.1.1 Regional differences introduced by perturbations

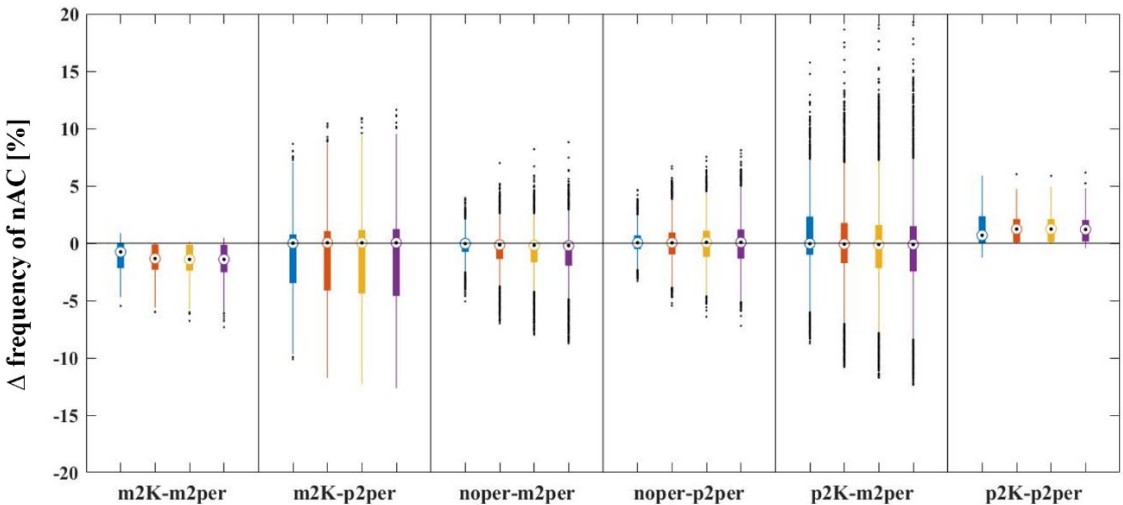

**Figure 2: Changes in frequency of non-atmospherically controlled (nAC) days in response to different combinations of temperature and moisture changes in the core-perturbation set. m2K denotes a cooling by 2K at the surface, p2K a warming of 2K at the surface, m2per denotes a dry of two times the scaling factor and p2per denotes a moistening by two times the respective scaling factor in the domain for different T-q scaling factors. Blue: 5% K$^{-1}$, orange: 7.74% K$^{-1}$, yellow: 8.81% K$^{-1}$, purple: 9.66% K$^{-1}$.**

In the core set, the modifications reach to approximately 500 hPa AGL. Moisture modifications followed as described above. The cases cover a range of different combinations of temperature and moisture modifications to estimate (1) modifications

with the same sign that represent changes following the observed positive correlations between T and q in Europe. Additionally, examining (2) the isolated effects of temperature and moisture, allows for the disentanglement of their impacts on the coupling strength as well as (3) modifications with opposing signs. The core set aimed at covering four possible combinations of differences in the climate conditions, namely, cooler and moister conditions, cooler and dryer conditions, warmer and moister conditions, as well as warmer and dryer conditions.






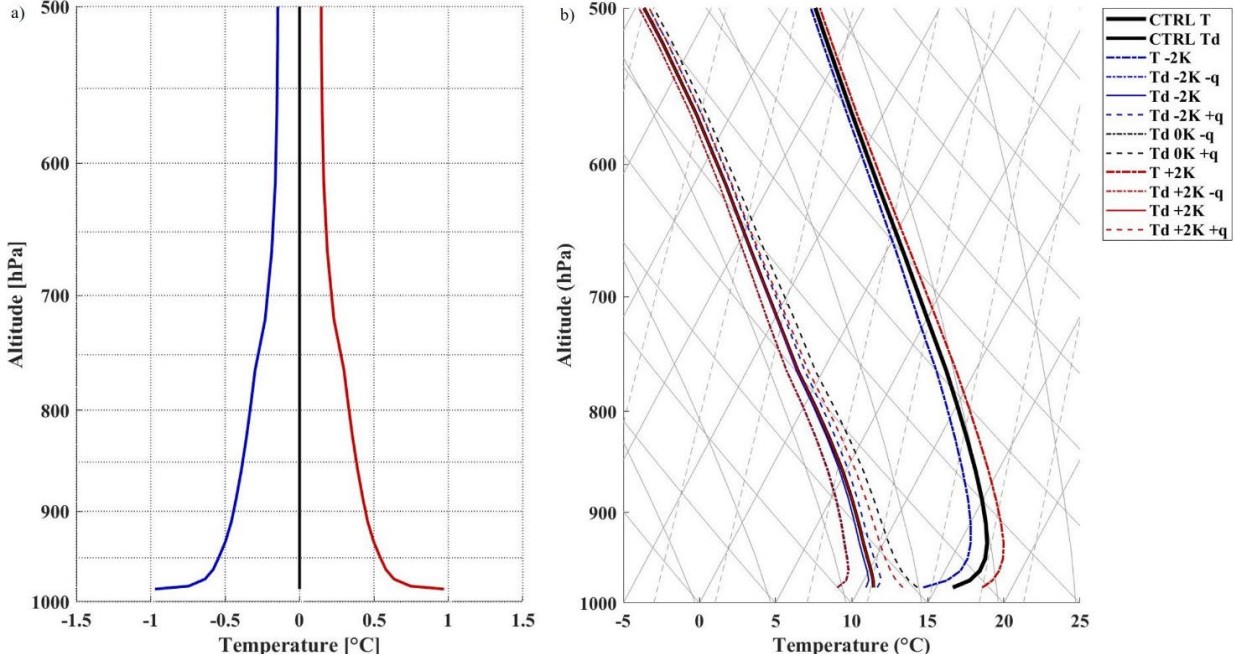

**Figure 3: Temperature perturbation factor derived using a simple linear regression model and extracting the coefficient of determination for each atmospheric layer (left). Profiles of temperature (T) and dew point temperature (Td) after perturbation (right). Red indicates warmer temperature, blue cooler temperatures and unchanged temperature is denoted in black. Dash-dotted lines indicate a reduction in moisture, solid lines unchanged moisture and dashed lines an increase in moisture.**

Previous observational and global model studies suggest that temperature and moisture are considerably positively correlated in most regions around the globe and trends lie around 7% change in moisture per K change in temperature, reflecting the Clausius-Clapeyron rate for increases in moisture and maintains a quasi-constant relative humidity (Bastin et al., 2019; Willett et al., 2010). In Europe, the scaling of moisture to temperature is slightly higher (section 2.3.2). In addition to the rates

described before, a rate of 5% $K^{-1}$ was tested to represent a change in moisture per K change in temperature below the Clausius-Clapeyron rate. Figure 2 depicts the divergence in frequency of nAC-days from the CTRL run with 2K warmer and cooler conditions for all land points. Impacts on the coupling strength and the pre-conditioning for the different feedback regimes have the same sign for each tested rate. A higher scaling of moisture with temperature - as observed in northern Europe - enhanced the effects on the coupling.

For the following analysis, we combined the rate of the northern hemisphere (8.81% $K^{-1}$) with 2K temperature changes at the land surface. Figure 3 shows the coefficient of determination used as basis for the perturbation over height as well as the temperature and dew point temperature profiles after perturbation. CTP and $HI_{low}$ changes are uniform throughout the domain. Their spatial patterns are largely maintained from the CTRL run, which are considered reasonable (Jach et al., 2020). When temperature and moisture perturbations have the same sign (e.g. warmer and moister), the sign of differences in nAC-days was

uniform through-out the domain (Fig. 4a,i). With cooler and dryer conditions reducing potential feedback days by about 5%, whereas warmer and moister conditions increase the frequency of nAC-days by 3-5%.





**Figure 4: Difference in the seasonal share of non-atmospherically controlled (nAC) days [%] from CTRL for each perturbation case of the core-set. The center image is the CTRL case modified after Jach et al. (2020) (their Fig. 4g). The columns denote the temperature change and the rows the relative change in moisture.**

Analyzing the cases with individual modifications in temperature and moisture are used to disentangle their respective impacts on different coupling variables. Isolated temperature changes primarily influence the coupling strength in northern Europe, where lower temperatures weaken the coupling over energy-limited regions – such as Scandinavia and over the Eastern European Plain. This happens in consequence of more early-morning profiles showing stable conditions. Conversely, a warming initiated a strengthening of the coupling (Fig. 4h). The impact was smaller in southern Europe, and it switched sign. Lower temperatures reduce the humidity deficit, and thus, decrease the amount of days during which a low atmospheric moisture content inhibits convective precipitation. Moisture modifications had a larger impact in the south of the domain. While dryer conditions were favorable for the occurrence of feedback days in the north, moister conditions were favorable in





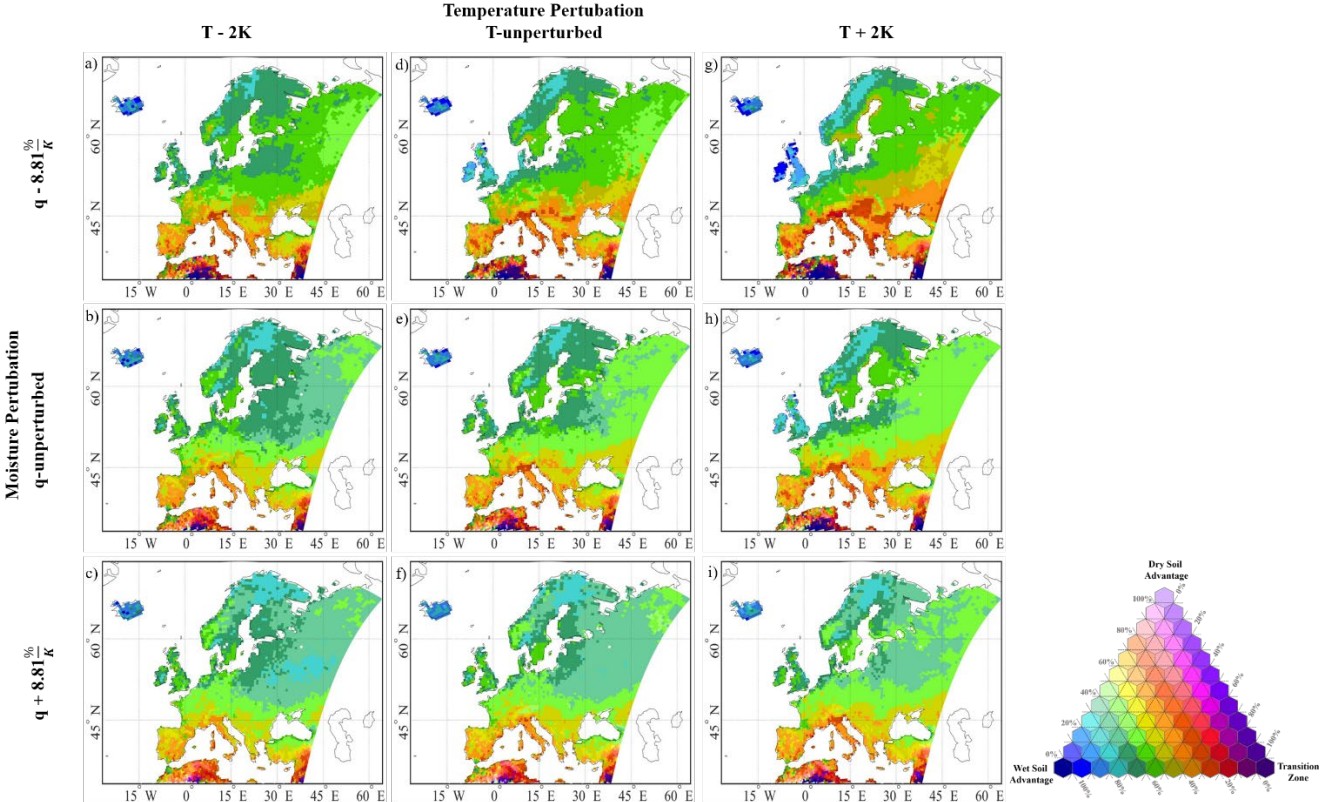

**Figure 5: Composition of the non-atmospherically controlled days comprising wet soil advantage, dry soil advantage and transition zone days for all core-perturbation cases. The columns denote the temperature change and the rows the relative change in moisture.**

the south. The same spatial patterns occurred when the implemented modifications differed in sign (Fig. 4c,g). Spatial patterns of impacts on the feedback variables are similar, and therefore, differences added up, leading to relatively high differences in the frequency of nAC-days (Fig. 4c,g) and their partitioning in wet and dry advantages (Fig. 5). Differences in the frequency of nAC-days reach up to 10% of the summer days. Nevertheless, following the argumentation that moisture scales positively with temperature, real-world temperature and moisture impacts are expected to counteract each other leading to weak net-

effects.

The partitioning of nAC-days experienced some small shifts of up to ±10% between the categories (Fig. 5). The predominance of the wet soil advantage in the north and of the transition zone around the Black Sea remains unaffected. The spatial patterns of changes in wet soil advantage days closely followed that in nAC-days in most perturbation cases. A change in the partitioning predominantly occurred between wet soil advantage and transition zone days. Dryer and warmer conditions

increased the frequency of transition zone days relative to the CTRL case, vice versa for moister and cooler conditions. Any perturbation case initiated a dominant dry soil advantage.





**Figure 6: Long-term classification of coupling regimes for the core-set perturbation cases. The columns denote a temperature change and the rows a change in moisture. The center image is the CTRL case and modified after Jach et al. (2020) (their Fig. 3a). The columns denote the temperature change and the rows the relative change in moisture.**

The impact on the long-term classification of coupling regimes does not reflect the changes in nAC-days and their partitioning in wet and dry advantages for convection (Fig. 6). Differences to the CTRL case mainly occur over Eastern Europe at the
edges of the feedback region, and the predominance for positive feedbacks remained unchanged also in the cases with strong changes in relative humidity. The perturbations initiated changes between wet soil advantage level 1 and 2, as well as transition zone level 1 and 2. None of the perturbation cases experienced to a considerable shift in location or predominant sign of feedbacks in the 30-year average (Fig. 5 and 6).



### 3.1.2 Sensitivity of the coupling to separated changes in temperature and moisture

This chapter further examines the relative importance of temperature versus moisture modifications for the variables CTP, $HI_{low}$, as well as the share of nAC-days, wet soil advantage, transition zone and dry soil advantage days in Europe. The sensitivity index as described in section 2.4 was used to estimate the magnitude of the control of temperature and moisture relative to each other for each variable throughout the domain.

The temperature and moisture perturbations changed CTP and $HI_{low}$ linearly. Differences in CTP, the stability of the
atmospheric layering, was almost solely controlled by modifications of the temperature, as indicated by a sensitivity index of -1 throughout the domain (not shown). In case of $HI_{low}$, the impacts of temperature and moisture modifications were of similar magnitude, though, moisture has a slightly higher impact, indicated by small but positive values. The magnitude of temperature and moisture controls on $HI_{low}$ becomes more equal in mountainous regions.

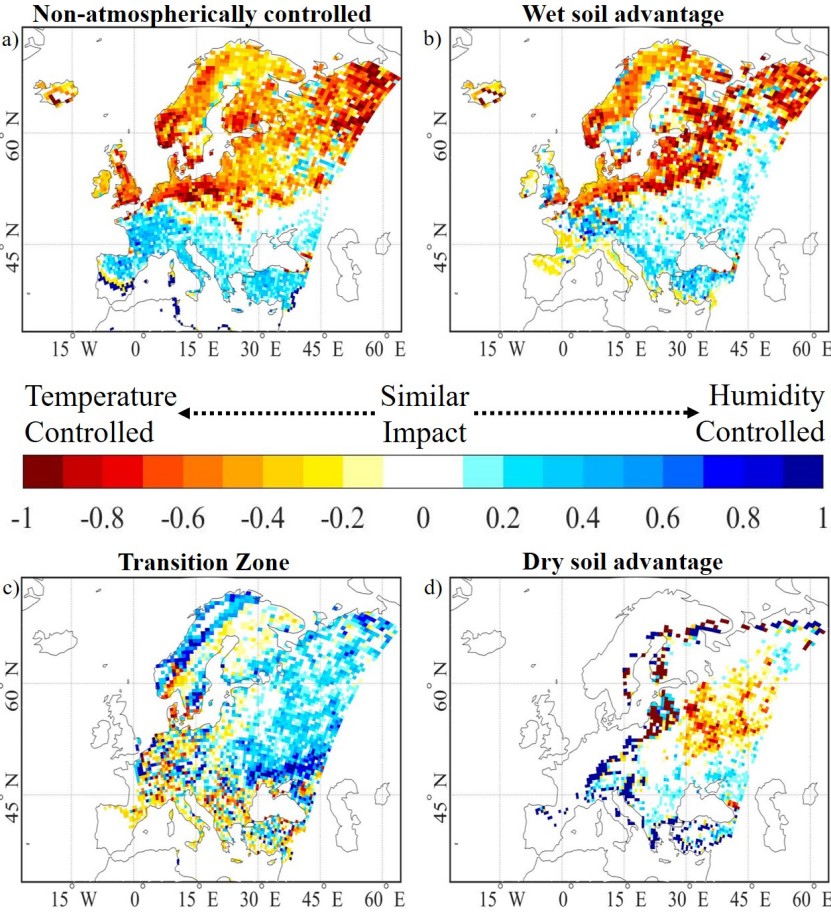

**Figure 7: Sensitivity score of a) non-atmospherically controlled days per summer season, b) wet soil advantage days per summer season, c) transition zone days per summer season and d) dry soil advantage days per summer season to changes individual modifications in the air temperature profile (-1 = totally temperature controlled) and specific humidity profiles (1 = totally humidity controlled).**





The sensitivity index for the share of nAC-days in summer showed a clear dipole pattern (Fig. 7a). In northern Europe, the coupling is rather impacted by temperature variations. Temperature controls the coupling by determining the stability of the atmosphere.

In southern Europe, moisture was the controlling factor, and little relative humidity in the low-level BL limits the occurrence of feedbacks in consequence of limited moisture availability for deep moist convection. The sensitivity index computed for the wet soil advantage showed a similar pattern. Hence, sensitivity of the coupling exhibited a regional dependency to temperature and moisture changes, which hints toward humidity- and energy-limited regimes controlling the coupling. The dry soil advantage rarely occurs, but its occurrence is rather controlled by temperature variations in northeastern Europe (Fig. 7d), and by moisture in southeastern Europe. The sensitivity of the transition zone shows a complete different pattern. The moisture modifications caused higher differences in the occurrence of transition zone days in the coupling hotspot, while towards the southwest temperature perturbations only had a higher impact (Fig. 7c).

### 3.1.3    Effects of changing temperature and moisture gradients

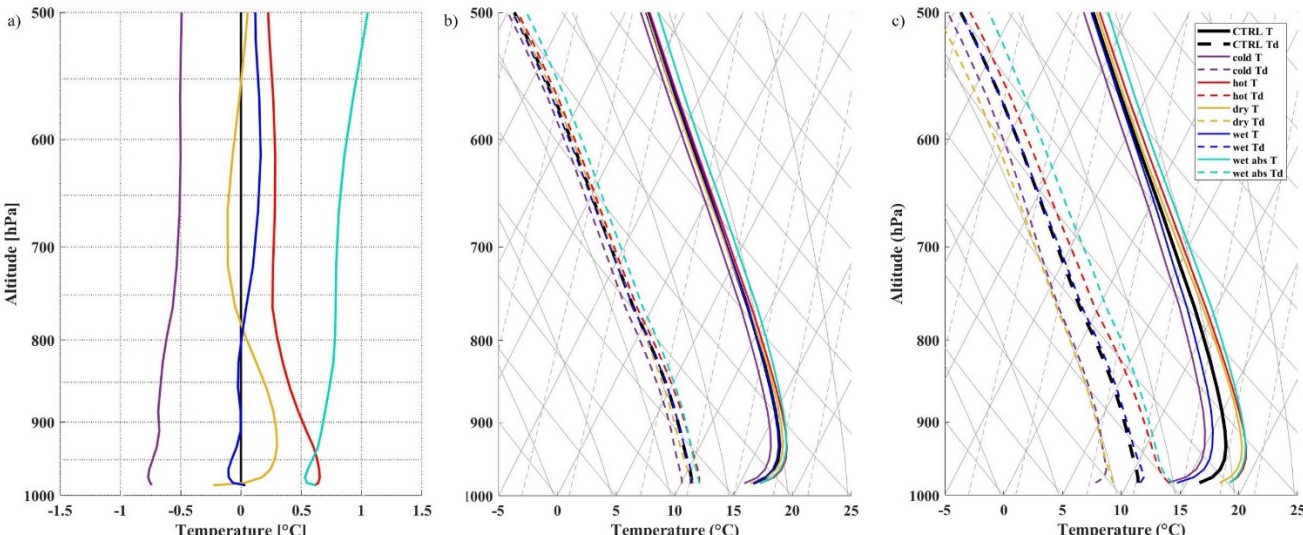

**Figure 8: a) divergence temperature (T) factors used to perturb daily model output, b) domain average of T and Td Profiles for the divergence T-factors, and c) their additional modifications with the core T-factor. purple: cold, red: hot, yellow: dry, blue: wet, turquoise: wet abs; Solid lines represent temperature and dashed lines represent dew point temperature.**

The following chapter deals with the analysis of how changes to steeper or less steep temperature and moisture gradients can influence the feedback classification and to compare how such differences can impact the result of the coupling metric. Figure 8 shows the divergence-factors for each case, as well as the resulting temperature and dew point temperature profiles of the lower BL. The cases chosen because of their moisture anomaly – namely the dry and the wet cases – the moisture-factor was



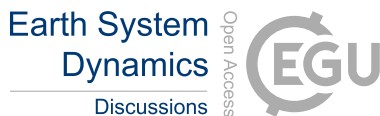

derived by multiplying the T-factor with -1 to derive moister condition in the wet and dryer conditions in the dry case. This was done to circumvent that in the dry case, a higher temperature would be associated with an increase in moisture (thus a moistening) of the BL with positive temperature-moisture relationship. As CTP is almost entirely controlled by the air temperature, this practice only affected $HI_{low}$.

We first investigate the impact of shifting the temperature and moisture gradients from the CTRL case using the divergence factors of the extreme years (see Section 2.3.1). The main impact concerns change in CTP, since this is an integrated variable. Changes in the temperature gradient move the lapse rate more toward the dry or moist adiabates, and hence influence the atmospheric stability. The hot and the dry- divergence factors increase the early-morning temperature gradients between 100-300 hPa above ground shifting it closer to the dry adiabat, but also enhance the surface inversion (Fig. 8). This causes an

increase in CTP, while the enhancement of the surface inversion, which is likely resulting in a higher convective inhibition, is not accounted for in the framework. In the other three cases (cold, wet, wet_abs) the temperature gradient is decreased between 100-300hPa AGL, consequently decreasing CTP (Fig. 9). The cases diverge in the mean temperature change among each other. Likewise, the temperature inversion decreases in the lower atmospheric layers (Fig. 8). Differences in $HI_{low}$ result from both temperature and moisture changes. However, $HI_{low}$ changes are small in most cases (Fig. 9), because temperature and

moisture change simultaneously, which leads to small changes in relative humidity. The only considerable exception is the dry case, where the T-factor was multiplied by -1. In this case, $HI_{low}$ increased by about 1°C.

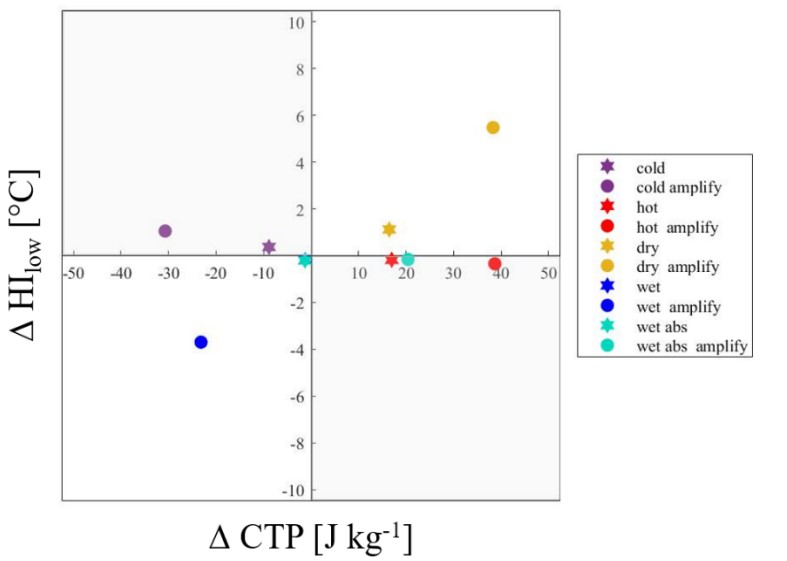

**Figure 9: Changes in convective triggering potential (CTP) and low-level humidity index ($HI_{low}$) due to the divergence factors.**



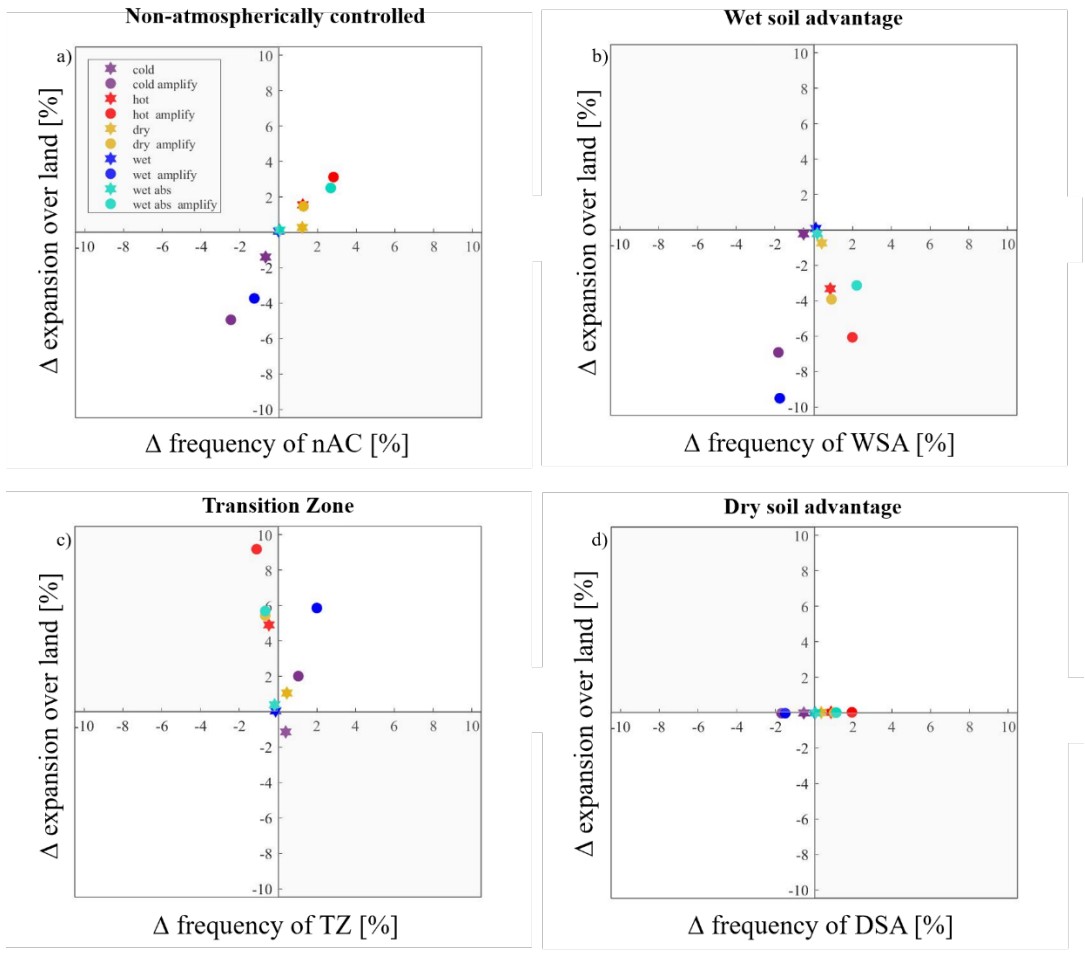

**Figure 10: Impacts of the divergence cases on the spatial expansion and the occurrence of the feedback categories in summer for a) non-atmospherically controlled (nAC) days, b) wet soil advantage (WSA), c) transition zone (TZ), and d) dry soil advantage (DSA). The x-axis depicts the changes in the average frequency of occurrence during summer and the y-axis shows changes in the fraction of land area covered by the respective coupling regime.**

The combination of temperature and moisture changes in each case determines the difference for the share of nAC-days (Fig. 10a). The effects are summarized in the following points:

- Hot case: Causes a higher temperature and temperature gradient between 100-300hPa AGL with corresponding changes in moisture. These lead to greater instability with a constant humidity deficit, which increases the expansion of the hotspot and the fraction of nAC-days within the hotspot.

- Dry case: Larger temperature gradient but less moisture in the atmosphere. A greater instability is combined with a higher humidity deficit, which jointly causes an increase in the fraction of nAC-days in summer in the hotspot, but

the area of the domain included in the hotspot remains unchanged. Higher humidity deficits reduce the coupling of



land surface and convection around the Black Sea, but increase the likelihood for convection triggering over wet soils in the north.

- Cold case: A combination of lower temperature, a decrease in the temperature gradient between 100-300hPa AGL, and moisture changes corresponding to 8.81% K$^{-1}$ lead to a reduction in the expansion of the hotspot region in the study area and a loss of nAC-days.

- Wet_abs case and wet case: temperature increases but a shallower temperature gradient with corresponding changes in moisture resulted in minor impacts on the coupling.

Further looking at the differences in the share of the coupling categories shows that the area in wet soil advantage shrinks in all divergence cases (Fig. 10b). Warmer temperatures strengthen the frequency of the wet soil advantage in the hotspot and
cooling weakens it. Days in the transition zone experience the opposite effect (Fig. 10c). However, all combinations of changes in the gradients lead to an expansion of the transition zone-labeled region over land. Though the dry soil advantage never becomes dominant , which can be seen in the unchanged expansion over land (Fig. 10d), temperature changes still influence the frequency of days during which negative feedbacks can occur. Similar to the wet soil advantage, higher temperatures increase the frequency of days in dry soil advantage during summer.

**3.2    Uncertainty of the coupling regimes**

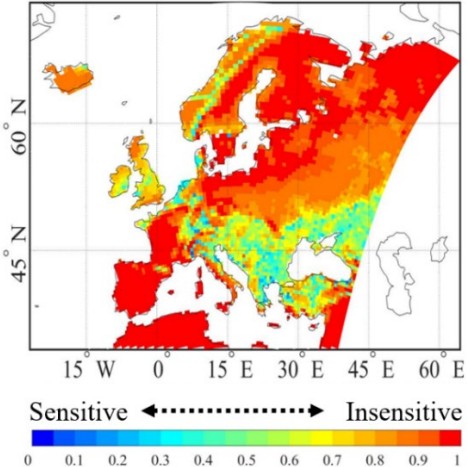

**Figure 11: Comparison of perturbation cases with CTRL from no perturbation case as CTRL. Red colors indicate that the feedback classification is sensitive to modifications in temperature and moisture, and greenish colors indicate that the feedback classification is insensitive to modifications in temperature and moisture.**

Here, we examine changes in the occurrence of the feedback classes during summer which is based on the daily classification (comp. Fig. 1a), and to which extent the long-term classification, indicating the dominance of a feedback class in a cell, reflects these changes. Under the assumption that the perturbation cases cover a reasonable spread in atmospheric

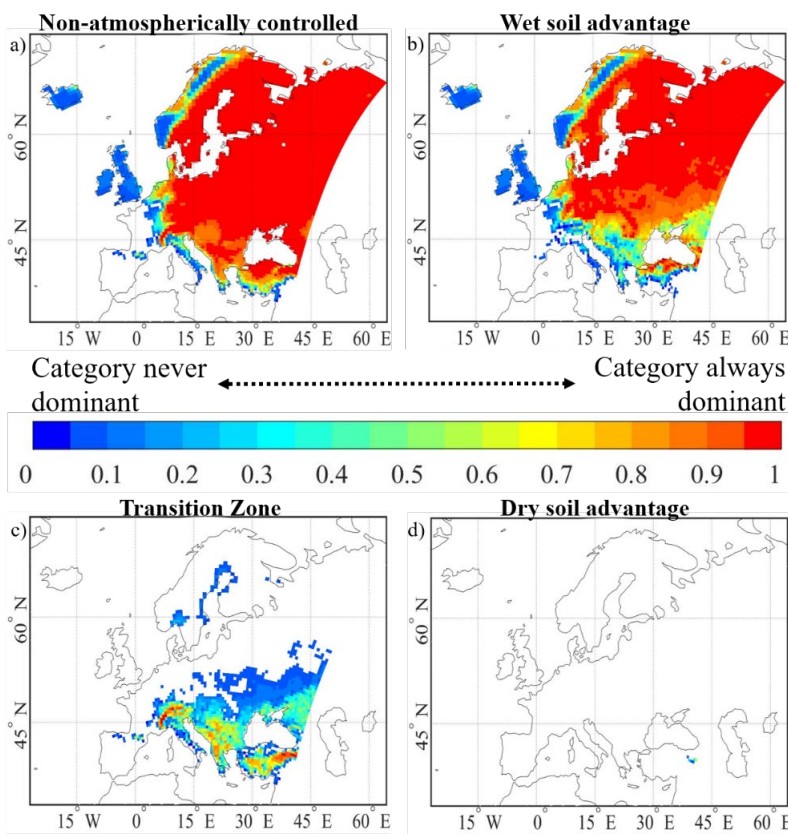

**Figure 12: Sensitivity of feedback categories of a) any feedback-category, b) classified as wet soil advantage level-1 or level-2, c) classified as transition zone level-1 or level-2, and d) classified as dry soil advantage level-1 or level-2**

temperature and moisture for the prevailing climate, it aims at understanding how sensitive the coupling strength, and the pre-dominant feedback respond to temperature and moisture differences within this spread. For this purpose, we first looked at the sensitivity of the long-term regime classification by determining the share of perturbation cases in which the feedback classification coincided with that of the CTRL case (Fig. 11). A high share as assessed with Eq. (3) indicated high agreement

in the long-term classification between the perturbation cases (red areas) and therefore low sensitivity, while green to blue colours indicate weak or no agreement of the perturbed feedback classifications with that of the CTRL case, and therefore, high sensitivity. Please not that no agreement also involves changes between a feedback regime in level-1 and level-2. We further quantified the frequency of occurrence of each feedback regime in the perturbation cases using Eq (4) to explore which feedback regimes are occurring in the different cases. The Iberian Peninsula, northern Africa and the northeast of Europe show

high agreement in the regime classification of all perturbation cases, and thus low sensitivity to temperature and moisture changes. Over the Iberian Peninsula and over northern Africa, the occurrence of precipitation is reliably in atmospheric control, whereas over northeastern Europe it was reliably in nAC (Fig. 12a). In the transition between these two regions occurred a belt, where the feedback regime changed on a regular basis. Thus, it appears to be sensitive to temperature and moisture changes. The absence of several feedback regimes suggests that Scandinavia, the British Isles and Mid-Europe, the question





is whether or not feedbacks occur, and when feedbacks occurred, wet soils are in favour (Fig. 12a,b). In southeastern Europe, between the Alps to around the Black Sea, summers are reliably in non-atmospherically control (Fig. 12a), but the dominant feedback regime switched between wet soil advantage and transition zone (Fig. 12 b,c). Some cells have an equal share of perturbation cases in wet soil advantage and transition zone. A dominant dry soil advantage occurs only in single cells and cases over Turkey.

Secondly, we explored differences regarding the occurrence of the different feedback classes within all summer days between the perturbation cases. This is based on the daily classification of the profiles in CTP-HI$_{low}$ space. The analysis of sensitivity in the long-term coupling regimes allows to distinguish five regions used for a spatial aggregation: (1) Pure nAC, where less than two perturbation cases changed the coupling regime maintaining nAC in nearly all cases, and (2) Pure AC, where less than two perturbation cases changed the coupling regime maintaining AC in nearly all cases. Further, there are three regions

with frequent switches (at least two cases) in the coupling regime. In region (3), the coupling regime changed between any AC class and the wet soil advantage, in (4) the change is between AC classes, the wet soil advantage and the transition zone, and in (5) the change is between the wet soil advantage and the transition zone. The cell remains in nAC in any of the perturbation cases. Figure 13 shows the distribution of summer days in the feedback classes for these regions and all cases. Figure 14 further adds sensitivity maps depicting the average dominance of each feedback regime relative to the other feedback classes and their

occurrence [d] in summer. Hatched areas denote that the number of days in the respective feedback regime varied considerably by more than 10% of the summer days between the perturbation cases.

In the Pure AC region, the perturbation cases' impact on the distribution is negligible. Dry AC days dominate, and modifications of temperature and moisture barely influenced the atmospheric pre-conditioning. Considerable variance in the occurrence of feedback days of in part more than 20% of the summer days occurred mainly in the hotspot region (Fig. 13 and

14d). In the Pure nAC region, the number of nAC-days ranged on spatial average between 19.2 and 28.5 days per season. The number of wet soil advantage days was relatively stable (ranged between 12.4 and 17.7 days), but the number of transition zone days varied in part considerably (between 4.3 and 11.8 days) with cases showing warming and great relative drying (p2K-m2per, dry amplification) having the most days in transition zone (Fig. 13a and 14b).

As indicated before, the classification is most variable in the WSA-TZ transition region. Similar to the Pure nAC region, the

number of nAC-days varied in spatial average between 15 and 26.1 days between the perturbation cases (Fig. 14d), but in contrast to the pure nAC region, the number of days in transition zone is relatively stable, and the number of days in wet soil advantage varied considerably (between 5.2 and 13.3 days) (Fig. 14a,b). The cases experiencing a strong reduction in relative humidity again show the strongest shifts in the average occurrence of feedback classes throughout the season, which can be seen in clearly less nAC-days and wet soil advantage days compared to the rest of the perturbation cases. In the AC-WSA

transition region, the number of nAC-days is at about the threshold of 10% distinguishing AC and nAC (compare Fig. 1b), and differences in the distribution of feedback classes are usually small. Only the cases experiencing warming combined with great reductions in relative humidity exhibit a considerable impact. These cases experience a clear increase in wet soil advantage days.





**Figure 13: Average distribution of the classes in the daily classification for all perturbation cases spatially aggregated in a) cells always in nAC, b) cells always in AC, c) cells in which the long-term classification frequently switched between AC and nAC, d) cells in which the long-term classification frequently switched between wet soil advantage (WSA) and transition zone (TZ).**





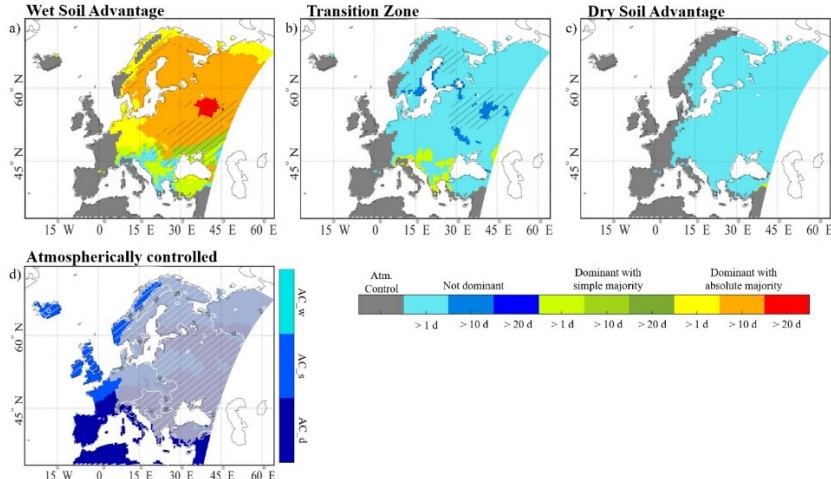

**Figure 14: Uncertainty maps of the non-atmospherically controlled classes: a) wet soil advantage, b) transition zone, c) dry soil advantage. The colours indicate whether a class is on average dominant in absolute or simple majority, or whether another class is dominant. The colour gradation denotes the average number of days. The hatching indicates that in these regions the variance in the number of days in this class is larger than 10%. Subplot d) indicates the dominant atmospherically controlled class the contours denote the maximum variance in the number of days in atmospheric control between the perturbation cases.**

The same analyses were also performed for perturbation cases with higher temperature modifications between ±5K and all combinations of moisture changes as done in the core perturbation set (not shown). This slightly enlarged the transition belt between AC and nAC, and increased the region where dominant wet soil advantage or transition zone can occur. Apart from that, the patterns for sensitive regions (Fig. 11) are substantially similar, and the absence of cells in dominant dry soil advantage remained unaffected.

**4    Discussion**

We perturbed daily temperature and moisture profiles around local sunrise of thirty summers from a regional climate simulation to examine the sensitivity of land-convection coupling strength to differences in the thermodynamic structure over Europe. The CTP-HI$_{low}$ framework was applied to each of eighteen perturbation cases grouped into two sets, on the one hand, to understand implications of warmer, cooler, moister or dryer atmospheric conditions for the coupling strength, and on the

other hand, to investigate the reliability of the strong coupling region's location and the predominant sign of feedbacks within the domain.

Studying spatial differences in the impacts of temperature and moisture changes reveals a north-south dipole in the coupling strength's sensitivity to changes in both variables indicated by a switch in the sign between the northern and the southern parts of the domain. Furthermore, temperature and moisture changes have contrary effects on the coupling strength throughout the

domain. This means that simultaneous increases or decreases, respectively, in temperature and moisture have small net-effects, and given that atmospheric temperature and moisture are strongly positively correlated in the northern hemisphere (e.g Willett



et al., 2010; Bastin et al., 2019), simultaneous changes of the same sign are considered most realistic. A strengthening of the coupling as a result of atmospheric warming is in line with the trend of stronger coupling in consequence of climate change over Europe. Seneviratne et al. (2006) showed the formation and expansion of a transitional region between wet and dry

climates over central and Eastern Europe in which strong L-A interactions can be expected. Dirmeyer et al. (2013) showed the trend of increasing coupling strength from a global perspective for both the land and the atmospheric segment.

Analyzing the relative importance of temperature versus moisture changes for the coupling strength within the domain suggests that the temperature control on the coupling strength is stronger in northern Europe, in particular that of feedback days in wet soil advantage (Fig. 7a,b), while moisture variations rather control the coupling strength in southern Europe. Please note that

the sign of changes in nAC-days and the feedback classes is not sensitive to the choice of the temperature-moisture scaling rate within a tested range of about $\pm$ 2% $K^{-1}$ around the Clausius-Clapeyron rate of 7 % $K^{-1}$ (Fig. 2). However, the rate does impact the magnitude of changes. In case of a rate below 7% $K^{-1}$, the impact of the respective perturbation cancels out in the more moisture controlled south. The areas of temperature and moisture control for nAC-days coincide with the energy- and moisture-limited regimes for evapotranspiration over Europe (Knist et al., 2017; Denissen et al., 2020; Seneviratne et al.,

2006). Our findings suggest that the energy and moisture limitations further propagate from the land segment of the coupling (connection between soil moisture and surface fluxes) to the atmospheric segment (connection between surface fluxes and boundary layer properties) along the local coupling process chain (Santanello et al., 2018).

Differences in the impacts of modified temperature and moisture gradients showed that the consideration of changes in the gradients can be as important for understanding differences in land-convection coupling as the temperature or moisture change

itself. It shows that increasing the temperature gradient, and hence destabilizing the atmosphere, usually increases the number of nAC-days, whereas shallower gradients reduce them. Thus, a warming signal propagating deeply through the atmospheric column (e.g. wet_abs, Fig. 9 and 10) leads to a smaller increase in the coupling strength than one that warms only the lower atmospheric levels resulting in a greater temperature gradient between 100-300 hPa AGL (hot). However, in the latter case, a stronger surface temperature inversion needs to be overcome by surface lifting, heating or moistening to enable buoyant lifting

and deep convection. This effect is not represented in the framework and might reduce the frequency of deep convection events, and hence, weaken the coupling again. Please note that the vertical resolution of the model (40 levels), limits the representation of details in the profiles, and a higher vertical resolution would provide a more accurate estimate of the temperature and moisture gradients (Wakefield et al., 2021). It is probable that the effects of altered gradients remain similar also with less details, though. Brogli et al. (2019) projected lapse-rate decreases in consequence of stronger upper-tropospheric

than surface warming over Europe by the end of the 21st century, and that the decreases are stronger over northern Europe than over the Mediterranean. Further research is necessary to quantify impacts on the coupling strength between surface wetness and convection triggering and future changes therein.

Finally, the reliability of the coupling hotspot as suggested by Jach et al. (2020) was analyzed, at first, by testing the sensitivity of the daily classification of atmospheric pre-conditioning in consequence of the perturbations, and secondly, by checking

whether and how frequently the dominance of a feedback advantage was changed over the thirty-year period. We have shown





that modifications of temperature and moisture cause considerable differences in both the occurrence of nAC-days and their partitioning in the different feedback classes over the strong coupling region throughout the summer season. However, this does not necessarily imply a change in the dominance of a feedback class. There are two regions in which the dominant feedback class is insensitive to changes in the atmospheric structure, wherefore the regime can be considered reliable. On the

one hand, the atmospherically controlled southwest and Atlantic coastal areas of Europe remain in atmospheric control in every perturbation case. Even considerable increases in low-level atmospheric relative humidity did not decrease the humidity deficit to a level in which local surface triggered deep convection can occur on a frequent basis. On the other hand, none of the perturbation cases reduced the coupling so that the strong positively coupled region over the Eastern European Plain disappeared. Thus, this region is considered a reliable hotspot region for positive feedbacks. Evidence for the location of the

hotspot is also found in Koster et al. (2004) or Seneviratne et al. (2006), who investigated hotspots of soil moisture-precipitation coupling in a global model ensemble.

Frequent changes in the coupling regime occur over parts of Scandinavia, Germany, and from the Alps to around the Black Sea. Regime changes are related to two effects or a combination of those. Firstly, the modifications frequently increase the number of nAC-days above the threshold to be considered nAC, and hence expand the size of the hotspot. This happens at the

border between the reliable AC and the strong coupling region. Differences among the perturbation cases are usually small, which suggests that the effect in reality is small. Secondly, the region from the Alps to around the Black Sea has always enough nAC-days to be considered nAC, but the dominant feedback class regularly shifts between the wet soil advantage and transition zone depending on the mean temperature and moisture. The number of wet soil advantage and transition zone days is fairly equal in this region. Differences in temperature and moisture control which class dominants, and hence, following their

definition, whether deep convection or shallow convection is more likely. This makes the region particularly interesting for future research on L-A feedback.

It has to be noted that the analysis focused on differences in the mean temperature and moisture including differences in the vertical distribution to approximate a potentially realistic spread in the atmospheric segment of L-A coupling strength for Europe. The horizontal and temporal distributions were maintained though differences in the temporal distribution are to be

expected in consequence of nonlinear feedback processes, when a change in the mean temperature and moisture occurs, which in turn can also impact the L-A coupling (Hirsch et al., 2014). Yet, a prediction of changes in the temporal distribution is complex and beyond what can be done with a perturbation factor. This suggests that further investigation is necessary to understand differences in the temporal distribution of temperature and moisture in the atmosphere and link them to L-A coupling to improve the understanding of modification in the coupling under changing climatic conditions.

**5    Summary**

By studying the sensitivity of the atmospheric segment of L-A coupling strength to perturbations in vertical temperature and moisture during 30 summers over Europe, we have shown that the atmospheric pre-conditioning is indeed sensitive to



temperature and moisture. However, no combination of temperature and moisture changes relocated or reshaped the coupling hotspot strongly over northeastern and eastern Europe. Differences in the frequency of occurrence of advantageous
atmospheric conditions for feedbacks of any kind suggest that uncertainty remains in the accuracy of the coupling strength itself, but stronger coupling relative to the rest of the domain is considered reliable there. Further research including the development of datasets usable for validation are required for an accurate approximation of the L-A coupling strength. Furthermore, the predominance of positive feedbacks, meaning convection is preferably triggered over wet soils, was preserved in all cases over the northern part of the coupling hotspot. Therefore, it is predestined for future studies on the impacts of
natural and deliberate land surface modifications on the local and regional climate as options for climate change mitigation, as an influence can be expected and the dominant response is certain. This is particularly interesting, in the light of rising temperatures and the related trend of strengthened L-A coupling under global warming (Dirmeyer et al., 2013; Seneviratne et al., 2006). In the southern part, the coupling classes wet soil advantage and transition zone have an equal share throughout summer, and temperature and moisture modifications cause a switch in the regime in several cases, implying uncertainty in
the dominant coupling regime. This makes the region particularly interesting for further studies on L-A coupling, because small changes in the atmospheric conditions might lead to a different atmospheric response. Additionally, the understanding and improved representation of these feedback processes in regional climate models are expected to reduce uncertainties in summer precipitation predictions in climate projections. Especially, the parameterization of convective precipitation has been shown to introduce uncertainties and more advanced triggering mechanisms for convection might lead to an improvement of
precipitation predictions (Chen et al., 2017).

Finally, process-based coupling studies still face a substantial lack of spatially comprehensive data covering the vertical structure of the BL on the regional scale, hence the reliance on model data. Efforts of creating a network of coordinated continuous long-term measurements such as the GLAFO initiative (Wulfmeyer et al., 2020) are required to close the gap and provide a validation basis for modeling-based studies. The modeling-based studies, in turn, are confronted with data storage
and computation limitations, which currently leads to the practice of storing 3D-fields only with a limited number of vertical levels. The trend of increasing complexity of atmospheric models, higher temporal and spatial resolutions, as well as spatial and temporal coverage of simulations strongly exacerbates storage limitations. Though, single model studies are limited in their generalizability as e.g. the choice of parameterizations or lateral boundary conditions cause uncertainty in coupling assessments, it is unlikely that comprehensive model ensemble studies will become feasible on the regional scale in the short
and medium term. Therefore, we consider this perturbation approach as a valuable alternative to study the sensitivity of the atmospheric segment of L-A coupling providing evidence for the location of a L-A coupling hotspot and a range for potential coupling strength under current climatic conditions.



## 6 Data availability

The data are available at the CERA database of the DKRZ (https://cera-
www.dkrz.de/WDCC/ui/cerasearch/entry?acronym=DKRZ_LTA_1140_ds00005.)

## 7 Author contribution

LJ performed the simulations and did the analysis. LJ and TS designed the analysis. LJ prepared the manuscript with contributions from all co-authors.

## 8 Competing interests

The authors declare that they have no conflict of interest

## 9 Acknowledgements

The research of this study was funded by the Anton and Petra Ehrmann-Stiftung Research Training Group "Water-People-Agriculture". We thank xx anonymous reviewers for their comments and helpful remarks on the manuscript. This work was completed in part with the CSL High-Performance Storage System provided by Computational Science Lab at the University
of Hohenheim, and we acknowledge support by the state of Baden-Württemberg through bwHPC.

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
