# Peer review of "Sensitivity of land-atmosphere coupling strength to changing atmospheric temperature and moisture over Europe"

_Earth System Dynamics, 2021_

## Author Comment (AC1)

Reviewer #2:

Summary/overall impression:

This article uses regional climate simulations to test the sensitivity of land-atmosphere coupling in Europe to changes in atmospheric moisture and temperature profiles, by applying a well-known land-atmosphere coupling metric. I genuinely enjoyed reading this article and feel that the results have important implications for considering the influence of L-A coupling in a changing climate. The article is well-organized and uses novel methods that are based upon previous studies to address the authors' hypotheses. I feel that this study will make a valuable contribution to the scientific literature. Moreover, the authors do an excellent job with the use of visuals to tell their story. As the authors' results are so clearly valuable, I have a few suggestions that I hope will help ensure the authors' main points are communicated clearly.

First of all, we would like to thank the reviewer for the supportive and constructive review. We hope we have satisfactorily addressed all suggestions and comments. Please find our responses below each corresponding comment.

Specific comments:

L24-25: I may suggest rearranging the first two sentences, leading off with what L-A coupling is. i.e. "Land-atmosphere coupling describes the covariability between land and atmospheric states, and plays a key role for understanding…." An additional suggestion here may be to specify which states in the climate system.

Response: We will revise the text accordingly.

L26: Schumacher et al. 2019 would be another relevant source for the influence of coupling on heat waves:

Schumacher, D. L., Keune, J., Van Heerwaarden, C. C., de Arellano, J. V. G., Teuling, A. J., & Miralles, D. G. (2019). Amplification of mega-heatwaves through heat torrents fuelled by upwind drought. *Nature Geoscience*, *12*(9), 712-717.

Response: We will add the citation.

L81: When you say. "The approach is based on they hypothesis…" it implies (to me) that this hypothesis was presented by numerous members of the broader scientific community and thus, supporting references should be provided. Though I am guessing you mean the hypothesis *you* are presenting specifically in this paper, in which case I might reword this to read "The approach is based on our hypothesis…"

Response: We agree with the point here and will make the change accordingly.

L84: It's not entirely clear what you mean by "the differences in the mean and vertical distribution." An extra sentence or two could improve clarity so that the reader knows what to expect in the rest of the analysis, especially if they are the type of reader that skips around sections and doesn't necessarily read the methods in depth. Are you meaning to say that you are considering how the mean changes, or how moisture and temperature deviate *from* the mean? Additionally, it may be helpful to clarify whether the change in vertical distribution going to be considered separately from your analysis of changes in the mean state (or deviation from the mean state) here.

Response: We agree that the wording is not entirely clear. We will edit and amend the text.

Section 2.1.1: This is well-organized, concise and easy to understand. Nice job.

Figure 1: I also really like how this figure is presented and summarizes your past work with respect to the metric you are using.

Figure 5: Once again… great use of visuals.

Response: Thank you these comments.

L343-346: Forgive me if this is beyond the scope of the current study, or if I missed something here. I find it interesting that the hot and dry divergence factors increase CTP, but also increase the surface inversion. While we generally associate higher CTP with dry soil advantage, could a greater inversion strength over wet soils also lead to more moisture buildup in the PBL, and thus a lowering of the LCL to the PBLH, that may also trigger convection? In this case, would we expect the CTP-HI metric to be sufficient for diagnosing coupling potential? Papers by Ek et al. (1994 and 2004) may be relevant to a discussion of impact on surface inversion here. You may ignore this comment if I'm missing the point.

Ek and L. Mahrt, 1994: Daytime Evolution of Relative Humidity at the Boundary Layer Top. Mon. Wea. Rev., 122, 2709–2721. doi: 10.1175/1520-0493

B. Ek and A. A. M. Holtslag, 2004: Influence of Soil Moisture on Boundary Layer Cloud Development. J. Hydrometeor, 5, 86–99. doi: 10.1175/1525-7541

Response: Thanks for this insightful and interesting comment. It is a nice line of thought to pick up.

First we would like to quote Ek and Mahrt (1994), who described a specific case in which subsidence "traps" surface moisture in a thin boundary layer, which increases the surface relative humidity. The timing of when the subsidence is overcome determines the development during the day. They state that "… *if this time* (to overcome the subsidence)

*is comparable to, or large compared to the period of mixed-layer development, then the main influence of the subsidence is the decrease the boundary-layer depth leading to smaller relative humidity at the boundary layer top compared to the case of no subsidence.*" (Ek and Mahrt, 1994, p2713). In this case, the subsidence/inversion would inhibit any coupling event.

Taking now the hot and the dry perturbation cases, they show on average stronger inversions as compared to the reference which inhibit boundary layer growth in the morning hours until they are overcome. Further, the temperature gradients above the inversion are stronger which hints to weaker stability, and this would support more rapid boundary layer growth after the inversion was overcome in these perturbation cases. The boundary layer growth could then support the mixing of the trapped moisture and potentially decrease the LCL to the PBLH. Given a wet soil, we would expect high surface evaporation moistening the boundary layer below the inversion in the morning hours, but also weak sensible heat fluxes and thus a weak "force" pushing against the inversion. Hence, weak BL growth would be expected in the morning hours over wet soils. This would hint to a longer timescale to overcome the inversion, which – following Ek and Mahrt (1994) – would lead to less clouds. Over dry soils, we would expect higher sensible heat fluxes and thus a higher probability to overcome the inversion and foster BL growth. Although, there is less moistening of the boundary layer from the surface expectable, it is more likely that the inversion is overcome and thus a coupling event could occur.

So, the case you referred to might indeed occur, but a dry coupling event seems more likely. However, whether a coupling event could occur over wet soil is expected to be strongly dependent on the inversion strength and whether there is still sufficient boundary layer heating to overcome the inversion early enough to enable the development of clouds. Also the entrainment flux is expected to play a role for the boundary layer development and the L-A coupling signal (van Heerwaarden et al., 2009). This information is not included in the framework without further extensions, as it neither considers the lowest 1000m of the BL, entrainment, nor the energy partitioning at the land surface. Hence, the CTP-HI$_{low}$ framework would not capture the effects of changes in inversion strength in any case.

van Heerwaarden, C. C., Vilà-Guerau de Arellano, J., Moene, A. F., and Holtslag, A. A. M.: Interactions between dry-air entrainment, surface evaporation and convective boundary-layer development: DRY-AIR ENTRAINMENT, SURFACE EVAPORATION AND CBL DEVELOPMENT, Q.J.R. Meteorol. Soc., 135, 1277–1291, https://doi.org/10.1002/qj.431, 2009.

L355-357: Since you are discussing the influence of temperature in the "hot case," it can be a bit confusing when you then say "fraction of nAC-days within the hotspot" as my mind

first thought hotspot in a literal, temperature sense. I would suggest changing this to read: "within the L-A coupling hotspot."

Response: Thank you for bringing this point up. We agree that the term hotspot can be misinterpreted in association with the hot case. We will revise the text according to your suggestion.

L360-362: Is this where CTP is increased, but so is the temperature inversion? So perhaps the likelihood of convective triggering over wet soils could be tied to the comments associated with L343-346 above?

Response: In the north, the increase in CTP - implying a destabilization of the atmosphere above the inversion - rather bumps stable (and eventually wet) atmospherically controlled days to the non-atmospherically controlled regime.

Following our argumentation above, the framework likely does not represent inversions, because the integration of CTP starts at 100hPa AGL. Hence, the process mentioned above would not be captured by the framework. Nevertheless, it still might occur, but we think that further analyses are necessary, which e.g. involve CIN to represent the inversion strength and additionally the surface fluxes as an indicator of whether the inversion can be overcome before the end of the mixed-layer development.

Broader comment regarding discussion: How might overall warming of the climate impact the length of the season in which we consider L-A coupling to be most influential? Your results imply that warming enhances coupling strength, so would that also mean that L-A coupling might be and important driver of hydroclimatic variability over a longer warm season? For example, instead of JJA, perhaps the "coupling season" would now be MJJAS?

Response: Dirmeyer et al. (2013) suggested an earlier springtime onset of L-A feedbacks over the US in the future. So given that warming enhances the coupling strength, a longer warm season might indeed also imply a prolonging of the "coupling season" to MJJAS over Europe. However, our analyses were not tailored to investigate this effect and do not provide enough evidence to give a sound answer to your questions.

Dirmeyer, P. A., Jin, Y., Singh, B., and Yan, X.: Trends in Land–Atmosphere Interactions from CMIP5 Simulations, 14, 829–849, https://doi.org/10.1175/JHM-D-12-0107.1, 2013.

L466-468: I wholeheartedly agree that we need more vertical resolution, everywhere, however, I do believe you can argue that while Wakefield et al. (2021) shows that vertical resolution is a limiting factor, you can still get representative estimates of the L-A coupling pre-conditioning even when vertical resolution is unfortunately limited. Therefore, I think you can use this reference to argue both points… your limitation in vertical resolution

introduces uncertainty, but that uncertainty is not so large that it substantially impacts the validity of your results.

Response: Thank you for bringing up this line or arguments. We will gladly pick it up in the discussion section.

L479: I'm not sure about the use of the word "reliable." My mind immediately jumps to an operational use of the word and thinking about model reliability. I do like that you say the feedback class is *insensitive* to changes though. Maybe "wherefore a consistent regime can be expected," if that's the message you are trying to convey.

Response: Thank you for this suggestion. We will revise the sentence.

Technical:

L123: Change "but maybe limit investigations" to "may limit investigations."

L142: Change "deep convection is inhibited by an inversion, only shallow clouds…" to "deep convection is inhibited by an inversion and only shallow clouds…" or separate into two sentences.

L382: Typo "Please not that…" should say. "Please note that…"

Response: We will incorporate all of your technical suggestions.

---

## Author Comment (AC2)

Reviewer #1:

The paper "Sensitivity of land-atmosphere coupling strength to perturbations of early-morning temperature and moisture profiles in the European summer" explores the uncertainty of classifying land-atmosphere interactions using the CTP-HI framework by systematically perturbing the temperature and moisture profiles from a regional climate model and then analyzing how the coupling classification changes. The paper is well written and has a logical organization that makes it easy to follow. The experiment design is interesting and the results provide new insights into the coupling between the land and the atmosphere for this particular model. Based on this, the paper is well suited for Earth System Dynamics and merits publication. Despite these positive aspects, the paper is not particularly clear in defining the larger research question and discussing the results in a way that is consistent with the work being done. Based on this, three suggestions for improvement are given below.

First, we would like to thank the reviewer for the time he invested in reviewing our manuscript, and for providing helpful and constructive comments. We hope we were able to address the issues raised appropriately in our responses, which are provided below each comment.

First, it is difficult to know how much to trust the results of this paper since the analysis is based on a regional climate model that may have its own set of biases that will skew the results from the coupling classification. As applied here, the CTP-HI classification is fixed and therefore, a model with a consistent bias in the atmospheric profiles will give skewed results. This climatological inconsistency in the CTP-HI framework for some data sets was shown in Ferguson and Wood (2011) and was the reason for developing a data set specific method of classifying the CTP-HI space (Roundy et al. 2013). One possible way of addressing this limitation is to compare the surface temperature and humidity from the model to observations. This would at least provide a means of assessing where the model is biased and may provide insights into the results such as why are there very few dry soil advantage days in the model (Figures 6 and 7). Regardless of what is done to address this, there needs to be a clearer discussion that the results in this paper are model specific and may or may not represent the real world.

Response: We agree with the reviewer that every climate model has its own set of biases and that there is potential for influencing the results. However, the goal of this work is to assess the extent at which temperature and moisture changes in the atmosphere might influence the land-atmosphere coupling regimes, and thus assess how the coupling signal changes rather that the classification of the coupling regimes themselves. The goal is not a verification of the coupling classification from the model.

Furthermore, while a comparison of the surface temperature field from the model with observations would of course be possible, such a comparison is challenging for surface humidity due to the lack of spatially comprehensive observations. The best option to evaluate the moisture fields is a comparison with reanalysis data such as the bias-corrected ERA5 reanalysis dataset (C3S, 2020).

To assess potential inconsistencies, we compared the temporal distributions of temperature and moisture from the model with reanalysis. As the frequency of occurrence of the coupling classes is rather linked to the temporal distribution of the temperature and moisture fields than to biases in the means, comparing the distributions is expected to be well suited to assess climatological inconsistencies. For this purpose, we applied two statistical measures on a cell-wise basis: a Z-statistic and the PDF skill score by Perkins et al. (2007). Both measures showed good agreement between the distributions of the model and the reanalysis data for both variables. The value of the Z-statistic remained below 2 for both variables throughout the entire domain, which means that the differences are statistically not significant. The PDF skill score has a value larger than 0.8 over most of the continent. Strongest discrepancies were found in the Mediterranean region, which is expected to be predominantly in atmospheric control.

We appreciate your insightful comment and agree that climatological inconsistency among datasets is a potential limitation which requires further space for discussion in the paper. This is why we will add a paragraph on this in the discussion section. However, as the focus of our study is to analyze changes in the coupling signal due to changes in moisture and temperature and not to verify the coupling signal in the model, we would not add an extra section with comparisons against reanalysis in the manuscript.

C3S: Near surface meteorological variables from 1979 to 2018 derived from bias-corrected reanalysis, https://doi.org/10.24381/CDS.20D54E34, 2020.

Perkins, S. E., Pitman, A. J., Holbrook, N. J., and McAneney, J.: Evaluation of the AR4 Climate Models' Simulated Daily Maximum Temperature, Minimum Temperature, and Precipitation over Australia Using Probability Density Functions, 20, 4356–4376, https://doi.org/10.1175/JCLI4253.1, 2007.

Second, on first reading the title and abstract, I thought this was more of a modeling study where the model was perturbed and then run like the original Findell paper. However, this work does not actually do any new model runs, nor does it actually look at coupled model processes within the model and could just as easily be applied to a reanalysis data set which would have the added benefit of having assimilated observations. This does not dimension the results but begs the question as to why a regional model run is used in the analysis as opposed to reanalysis? Why not do both and compare them?

Response: Thank you for your comment. The main reason for using the regional climate model run was to maintain consistency with the investigations of Jach et al. (2020) in which additional model simulations with modified land cover were analyzed and which is referred to throughout. This was meant to provide a comprehensive picture on the coupling strength and factors at the land surface and in the atmosphere which potentially influence the long-term coupling signal. Further, we intend to apply this methodology to model runs of future periods for which no reanalysis data exist.

Since we wanted to focus primarily on the changes in the coupling signal due to modifications in temperature and moisture in this work, we are convinced that the results are meaningful also without the benefit of assimilated observations as given by reanalysis data. Nevertheless, we agree that a comparison of the model results with results from reanalysis data would be interesting for estimating uncertainty coming from the climatological inconsistencies between datasets as you raised in your first comment. We think this is an interesting option for future analysis which we will mention in the conclusion, but it is beyond the scope of this paper.

At a minimum, revising the title and abstract so that it better reflects the work done would be beneficial. In my opinion, this work is interesting because it is answering the question of what happens to the coupling if there is a change in temperature or moisture? It would be great to see the title and abstract reflect this.

Response: We understand your point and agree with you. We will revise the title and the abstract so that the work done is better reflected in there.

Third, on a whole the results are fairly predictable in that if you change the temperature and humidity profiles then you will change the calculated CTP-HI, which will then change coupling classification for that particular day. This means that areas that will be most affected will be those that lie on the boundaries between the strict classification thresholds. So what is really being analyzed in this work is what regions are most often on the boarder of the classification regimes and what kind of perturbations will bump them into the other regime. This is not to say that work is not meaningful, but I think it would greatly improve the paper by discussing this simple idea extensively in the introduction to help better setup the results.

Response: It is indeed true that the perturbations are meant to test whether, where and under which conditions they modify -or as you say bump- the coupling classification into another class. This not apparent from the regime classification of the model output only, and thus needs to be characterized and quantified based on the model output and its perturbations. Assuming that the classification is accurate enough, the coupling is vulnerable to changes in temperature and moisture in a region in which the classification is regularly bumped into another class. This is because the atmospheric preconditioning

is at the thresholds between the different classes and a bump implies that the likelihood for a certain response in the atmosphere changes from one to another. We will add a paragraph in the introduction and broached it of in the discussion.

In addition, below are several minor suggestions for improving the paper. Lines 74-75: The CTP-HI framework has been applied using satellite data and has given reasonable results (Roundy and Santanello 2017).

Response: Thank you for pointing out this study to us. We apologize that we omitted it and will revise the corresponding paragraphs.

Line 123: Consider revising to "but may limit the investigation of pre-conditioning"

Response: We will adopt this suggestion.

Lines 310 and 332: There are a couple instances of using the word chapter in the paper. For this kind of paper, "section" would be better.

Response: Thank you for mentioning that. We will change all occurrences of "chapter" to "section".

Line 352: The figure caption needs more detail here. Is this the average for the entire domain or just part of it?

Response: To achieve the factors, we averaged over the entire domain. We will amend the caption and give it more details, as well as clarify it in the text.

Line 386: Precipitation is not really validated in this work. This may be true if one assumes that the Findell et al. framework holds for the model used in this study, but no analysis is given to show this. It is probably best to avoid making the jump to precipitation and just stick with the classification.

Response: We agree with your comment and also didn't mean to imply that precipitation or the outcome in form of a traceable coupling event was validated. We will revise the paragraph.

References
Ferguson, C. R., and E. F. Wood, 2011: Observed Land-Atmosphere Coupling from Satellite Remote Sensing and Reanalysis. J. Hydrometeorol., 12, 1221–1254, https://doi.org/10.1175/2011jhm1380.1.
Roundy, J. K., and J. A. Santanello, 2017: Utility of Satellite Remote Sensing for Land-Atmosphere Coupling and Drought Metrics. J. Hydrometeorol., 18, 863–877,

https://doi.org/10.1175/JHM-D-16-0171.1.

Roundy, J.K., C. R. Ferguson, and E. F. Wood, 2013: Temporal Variability of Land–Atmosphere Coupling and Its Implications for Drought over the Southeast United States. J. Hydrometeorol., 14, 622–635, https://doi.org/10.1175/JHM-D-12-090.1.

---

## Author Response (AR1)

Reviewer #1:

The paper "Sensitivity of land-atmosphere coupling strength to perturbations of early-morning temperature and moisture profiles in the European summer" explores the uncertainty of classifying land-atmosphere interactions using the CTP-HI framework by systematically perturbing the temperature and moisture profiles from a regional climate model and then analyzing how the coupling classification changes. The paper is well written and has a logical organization that makes it easy to follow. The experiment design is interesting and the results provide new insights into the coupling between the land and the atmosphere for this particular model. Based on this, the paper is well suited for Earth System Dynamics and merits publication. Despite these positive aspects, the paper is not particularly clear in defining the larger research question and discussing the results in a way that is consistent with the work being done. Based on this, three suggestions for improvement are given below.

We would like to thank reviewer #1 for the time he invested in reviewing our manuscript, and for providing helpful and constructive comments. We assume we were able to address the issues raised appropriately in our responses, which are provided below each comment.

First, it is difficult to know how much to trust the results of this paper since the analysis is based on a regional climate model that may have its own set of biases that will skew the results from the coupling classification. As applied here, the CTP-HI classification is fixed and therefore, a model with a consistent bias in the atmospheric profiles will give skewed results. This climatological inconsistency in the CTP-HI framework for some data sets was shown in Ferguson and Wood (2011) and was the reason for developing a data set specific method of classifying the CTP-HI space (Roundy et al. 2013). One possible way of addressing this limitation is to compare the surface temperature and humidity from the model to observations. This would at least provide a means of assessing where the model is biased and may provide insights into the results such as why are there very few dry soil advantage days in the model (Figures 6 and 7). Regardless of what is done to address this, there needs to be a clearer discussion that the results in this paper are model specific and may or may not represent the real world.

We agree with the reviewer that every climate model has its own set of biases and that there is potential for influencing the results. While a comparison of the surface temperature field from the model with observations would be possible, such a comparison is challenging for surface humidity due to the lack of spatially comprehensive observations. The best option to evaluate the moisture fields is a comparison with reanalysis data such as the bias-corrected ERA5 reanalysis dataset (C3S, 2020). Therefore, we added corresponding results in this work.

To assess potential uncertainties arising from climatological inconsistencies in the chosen data set, we compared the temporal distributions of temperature and moisture from the model with reanalysis in addition to the bias. As the frequency of occurrence of the nAC-days and their partitioning in wet soil advantage, dry soil advantage and transition zone is linked to the temporal distribution of temperature and moisture fields, comparing these is expected to indicate whether the model represents the relative frequency of the coupling classes sufficiently well.

We applied two statistical measures on a cell-wise basis: a Z-statistic and the PDF skill score by Perkins et al. (2007). Both measures showed good agreement between the distributions of the model and the reanalysis data for both variables, particularly over Northern and Central Europe. The Z-statistic gave a value below 2 for temperature as well as moisture throughout the entire domain, which means that the differences are statistically insignificant. The PDF skill score has a value larger than 0.8 over most of the continent. Both measures also reveal weaker agreement over the Mediterranean region, which is expected to be predominantly in atmospheric control. Nevertheless, the model has a dry and warm bias over southern Europe and a cold bias over Eastern Europe, which likely influences the occurrence of the coupling classes.

We appreciate your insightful comment and agree that climatological inconsistency among datasets is a potential limitation which requires further space for discussion in the paper. Therefore, we added these analyses in section 3.1 including the new Fig. 2 showing the comparisons between model and ERA5 reanalysis data and in the discussion. Particularly, we included the following extensions in the text:

[revised manuscript text omitted]

Second, on first reading the title and abstract, I thought this was more of a modeling study where the model was perturbed and then run like the original Findell paper. However, this work does not actually do any new model runs, nor does it actually look at coupled model processes within the model and could just as easily be applied to a reanalysis data set which would have the added benefit of having assimilated observations. This does not dimension the results but begs the question as to why a regional model run is used in the analysis as opposed to reanalysis? Why not do both and compare them?

Thank you for your comment. The main reason for using the regional climate model run was to maintain consistency with the investigations of Jach et al. (2020), in which additional model simulations with modified land cover were analyzed and to which is referred to throughout the manuscript. The current study was meant to contribute to a comprehensive picture of the coupling strength involving the analysis of factors at the land surface and in the atmosphere, which potentially influence the long-term coupling signal.

We agree that a comparison of the model results with results from reanalysis data would be interesting for further quantifying the uncertainty coming from the climatological inconsistencies between datasets as you raised in your first comment. We think this is an interesting option for future analysis which we have mentioned it in the summary section, but it is beyond the scope of this paper. As mentioned in the response to the first comment, we have statistically compared the modelled near-surface temperature and specific humidity with that of a bias-corrected ERA5 reanalysis dataset which revealed good agreement in the temporal distributions of both data sets, particularly in the hotspot region. Since we wanted to focus primarily on the changes in the coupling signal due to modifications in temperature and moisture in this work and based on the findings of the statistical comparisons, we are convinced that the results are meaningful also without the benefit of assimilated observations as given by reanalysis data.

We added the results of the statistical analysis in section 3.1, added Fig. 2 and amended a paragraph on the comparisons in the discussion (Please find the new section 3.1 as well as the paragraph of the discussion in our response to the first comment above), Further, we added the following sentence in the summary:

> Summary L574-576: *"Further research including the development of datasets usable for validation or the analysis of L-A coupling in the most recent reanalysis datasets are required for refined approximations of the L-A coupling strength."*

At a minimum, revising the title and abstract so that it better reflects the work done would be beneficial. In my opinion, this work is interesting because it is answering the question of what happens to the coupling if there is a change in temperature or moisture? It would be great to see the title and abstract reflect this.

We understand your point and agree with you. We have revised the title and the abstract so that the work done is better reflected in there.

> Title: *"Sensitivity of land-atmosphere coupling strength to changing atmospheric temperature and moisture over Europe"*

> Abstract L7-25: *"The quantification of land-atmosphere coupling strength is still challenging, particularly in the atmospheric segment of the local coupling process chain. This is in part caused by a lack of spatially comprehensive observations of atmospheric temperature and specific humidity which form the verification basis for the common process-based coupling metrics. In this study, we aim at investigating where uncertainty in the atmospheric temperature and moisture affect the land-atmosphere coupling strength over Europe, and how changes in the mean temperature and moisture, as well as their vertical gradients influence the coupling. For this purpose, we implemented systematic a-posteriori modifications to the temperature and moisture fields from a regional climate simulation to create a spread in the atmospheric conditions. Afterwards, the process-based coupling metric 'convective triggering potential – low-level humidity index framework was applied to each modification case.*

> *Comparing all modification cases to the unmodified control case revealed that a strong coupling hotspot region in north-eastern Europe was insensitive to temperature and moisture changes, although the number of potential feedback days varied by up to 20 days per summer season. The predominance of positive feedbacks remained unchanged in the northern part of the hotspot, alike none of the modifications changed the frequent inhibition of feedbacks due to dry conditions in the atmosphere over the Mediterranean and the Iberian Peninsula. However, in the southern hotspot region in the north of the Black*

*Sea, the dominant feedback class frequently switched between wet soil advantage and transition zone. Thus, both the coupling strength and the predominant sign of feedbacks were sensitive to changes in temperature and moisture in this region. This implies not only uncertainty in the quantification of land-atmosphere coupling strength but also the potential that climate change induced temperature and moisture changes considerably impact the climate there, because they also change the predominant atmospheric response to land surface wetness."*

Third, on a whole the results are fairly predictable in that if you change the temperature and humidity profiles then you will change the calculated CTP-HI, which will then change coupling classification for that particular day. This means that areas that will be most affected will be those that lie on the boundaries between the strict classification thresholds. So what is really being analyzed in this work is what regions are most often on the boarder of the classification regimes and what kind of perturbations will bump them into the other regime. This is not to say that work is not meaningful, but I think it would greatly improve the paper by discussing this simple idea extensively in the introduction to help better setup the results.

It is indeed true that the modifications were meant to test whether, where and under which conditions they modify the coupling classification into another class. It is the central idea of the study to identify these regions, because under the assumption that the classification is accurate enough, regular changes in the classification imply that the coupling is vulnerable to changes in temperature and moisture in this region. This is because the atmospheric preconditioning is at the thresholds between the different classes and a change implies that the likelihood for a certain response in the atmosphere changes from one to another. This is not apparent from the regime classification of the model output only, and thus needs to be characterized and quantified based on the model output and its modifications. We revised and amended the introduction and broached it of in the discussion.

> Introduction L83-91: *"To study how sensitive the atmospheric segment of L-A coupling strength responds to differences in the atmospheric pre-conditioning, we developed an approach with which the temperature and moisture output fields from a regional climate model run were modified after the simulation and before applying the CTP-HI$_{low}$ framework. The modifications are expected to change the pre-conditioning and thus potentially the coupling classification. First of all, frequent changes in the classification show that it lies at the boundaries of different classes. However, assuming that the classification framework is accurate enough, frequent changes also reveal that the expectable coupling signal remains uncertain. This is shown as changes in the*

*atmospheric conditions in a presumably realistic range for the current climate could initiate different atmospheric responses such as triggering deep, shallow or no convection in different cases in the same region. Furthermore, it indicates a sensitivity of the coupling to changes in the atmosphere e.g. arising from climate change or changes at the land surface."*

Discussion L472-474: *"Analyses of the latter base on the idea that regions lying at the boundaries of two or more categories are particularly sensitive to changes in the atmosphere, as small changes in the pre-conditioning could initiate a different atmospheric response to surface wetness conditions. "*

In addition, below are several minor suggestions for improving the paper. Lines 74-75: The CTP-HI framework has been applied using satellite data and has given reasonable results (Roundy and Santanello 2017).

Thank you for pointing out this study to us. We revised the corresponding lines of the revised manuscript to:

L78-80: *"Other observational products such satellite-based profile data were already successfully used to apply the CTP-HI$_{low}$ framework on (Roundy and Santanello, 2017), although they often have coarse vertical resolutions (Wulfmeyer et al., 2015)."*

Line 123: Consider revising to "but may limit the investigation of pre-conditioning"

We have adopted this suggestion. The sentence in the revised document is now:

L136-137: *"The pressure height estimates are valid for Europe, but may limit the investigation of pre-conditioning in hot and arid regions, where the BL usually grows to higher altitudes throughout the day."*

Lines 310 and 332: There are a couple instances of using the word chapter in the paper. For this kind of paper, "section" would be better.

Thank you for mentioning that. We have changed all occurrences of "chapter" to "section".

Line 352: The figure caption needs more detail here. Is this the average for the entire domain or just part of it?

To achieve the factors, we averaged over the entire domain. We have amended the caption and gave it more details, as well as clarified it in the text.

Caption: *"Figure 9: a) Divergence temperature (T) factors derived from differences of the domain average temperature profiles of the corresponding summers to the 30-year mean (Tab. 2) which were used to modify daily model output, b) domain average of T and Td Profiles for the divergence T-factors,*

*and c) their additional modifications with the core T-factor. purple: cold, red: hot, yellow: dry, blue: wet, turquoise: wet abs; Solid lines represent temperature and dashed lines represent dew point temperature."*

*L370-372: "Figure 9 shows the divergence-factors for each case which were derived from the temperature difference of the corresponding summer (Tab. 2) from the climatological mean temperature averaged over the domain."*

Line 386: Precipitation is not really validated in this work. This may be true if one assumes that the Findell et al. framework holds for the model used in this study, but no analysis is given to show this. It is probably best to avoid making the jump to precipitation and just stick with the classification.

We agree with your comment and also didn't mean to imply that precipitation or the outcome in form of a traceable coupling event was validated. We have revised the paragraph to:

L 422-425: *"The Iberian Peninsula, northern Africa and the northeast of Europe show high agreement in the regime classification of all modification cases, and thus low sensitivity to temperature and moisture changes. Over the Iberian Peninsula and over northern Africa, the dry atmospheric controlled regime reliably predominated in all cases, whereas over north-eastern Europe was reliably classified in the nAC coupling regimes (Fig. 13a)."*

"References

Ferguson, C. R., and E. F. Wood, 2011: Observed Land-Atmosphere Coupling from Satellite Remote Sensing and Reanalysis. J. Hydrometeorol., 12, 1221–1254, https://doi.org/10.1175/2011jhm1380.1.

Roundy, J. K., and J. A. Santanello, 2017: Utility of Satellite Remote Sensing for Land-Atmosphere Coupling and Drought Metrics. J. Hydrometeorol., 18, 863–877, https://doi.org/10.1175/JHM-D-16-0171.1.

Roundy, J.K., C. R. Ferguson, and E. F. Wood, 2013: Temporal Variability of Land–Atmosphere Coupling and Its Implications for Drought over the Southeast United States. J. Hydrometeorol., 14, 622–635, https://doi.org/10.1175/JHM-D-12-090.1.

Reviewer #2:

Summary/overall impression:

This article uses regional climate simulations to test the sensitivity of land-atmosphere coupling in Europe to changes in atmospheric moisture and temperature profiles, by applying a well-known land-atmosphere coupling metric. I genuinely enjoyed reading this article and feel that the results have important implications for considering the influence of L-A coupling in a changing climate. The article is well-organized and uses novel methods that are based upon previous studies to address the authors' hypotheses. I feel that this study will make a valuable contribution to the scientific literature. Moreover, the authors do an excellent job with the use of visuals to tell their story. As the authors' results are so clearly valuable, I have a few suggestions that I hope will help ensure the authors' main points are communicated clearly.

We would like to thank reviewer #2 for the supportive and constructive review. We assume we have satisfactorily addressed all suggestions and comments. Please find our responses below each corresponding comment.

Specific comments:

L24-25: I may suggest rearranging the first two sentences, leading off with what L-A coupling is. i.e. "Land-atmosphere coupling describes the covariability between land and atmospheric states, and plays a key role for understanding...." An additional suggestion here may be to specify which states in the climate system.

We have revised the text accordingly. The first sentence now reads as follows:

L27-28: *"Land-atmosphere (L-A) coupling describes the covariability between the land and atmospheric states, and plays a key role for understanding states in the climate system such as the evolution of ABL temperatures and humidities."*

L26: Schumacher et al. 2019 would be another relevant source for the influence of coupling on heat waves:

Schumacher, D. L., Keune, J., Van Heerwaarden, C. C., de Arellano, J. V. G., Teuling, A. J., & Miralles, D. G. (2019). Amplification of mega-heatwaves through heat torrents fuelled by upwind drought. *Nature Geoscience*, *12*(9), 712-717.

We have added the citation.

L81: When you say. "The approach is based on they hypothesis…" it implies (to me) that this hypothesis was presented by numerous members of the broader scientific community

and thus, supporting references should be provided. Though I am guessing you mean the hypothesis *you* are presenting specifically in this paper, in which case I might reword this to read "The approach is based on our hypothesis…"

We agree with the point here and have made the change accordingly.

L84: It's not entirely clear what you mean by "the differences in the mean and vertical distribution." An extra sentence or two could improve clarity so that the reader knows what to expect in the rest of the analysis, especially if they are the type of reader that skips around sections and doesn't necessarily read the methods in depth. Are you meaning to say that you are considering how the mean changes, or how moisture and temperature deviate *from* the mean? Additionally, it may be helpful to clarify whether the change in vertical distribution going to be considered separately from your analysis of changes in the mean state (or deviation from the mean state) here.

We agree that the wording is not entirely clear. We have edited and amended the text:

L95-98: *"Here, we focus on the impact of differences in the mean states and the vertical gradients of temperature and specific humidity in the perturbation cases compared to the CTRL. For this purpose, we have set up two sets of posterior modification cases, one targeting the analysis of differences in the mean state and one the analysis of differences in the vertical gradients."*

Section 2.1.1: This is well-organized, concise and easy to understand. Nice job.

Figure 1: I also really like how this figure is presented and summarizes your past work with respect to the metric you are using.

Figure 5: Once again… great use of visuals.

Thank you for these comments.

L343-346: Forgive me if this is beyond the scope of the current study, or if I missed something here. I find it interesting that the hot and dry divergence factors increase CTP, but also increase the surface inversion. While we generally associate higher CTP with dry soil advantage, could a greater inversion strength over wet soils also lead to more moisture buildup in the PBL, and thus a lowering of the LCL to the PBLH, that may also trigger convection? In this case, would we expect the CTP-HI metric to be sufficient for diagnosing coupling potential? Papers by Ek et al. (1994 and 2004) may be relevant to a discussion of impact on surface inversion here. You may ignore this comment if I'm missing the point.

Ek and L. Mahrt, 1994: Daytime Evolution of Relative Humidity at the Boundary Layer Top. Mon. Wea. Rev., 122, 2709–2721. doi: 10.1175/1520-0493

B. Ek and A. A. M. Holtslag, 2004: Influence of Soil Moisture on Boundary Layer Cloud Development. J. Hydrometeor, 5, 86–99. doi: 10.1175/1525-7541

Thanks for this insightful and interesting comment. It is a nice line of thought to pick up.

First, we would like to quote Ek and Mahrt (1994), who described a specific case in which subsidence "traps" surface moisture in a thin boundary layer, which increases the surface relative humidity. The timing of when the subsidence is overcome determines the development during the day. They state that "… *if this time* (to overcome the subsidence) *is comparable to, or large compared to the period of mixed-layer development, then the main influence of the subsidence is to decrease the boundary-layer depth leading to smaller relative humidity at the boundary layer top compared to the case of no subsidence."* (Ek and Mahrt, 1994, p2713). In this case, the subsidence/inversion would inhibit any coupling event.

Taking now the hot and the dry perturbation cases, they show on average stronger inversions as compared to the reference, which inhibit boundary layer growth in the morning hours until they are overcome. Further, the temperature gradients above the inversion are stronger which hints to weaker stability, and this would support more rapid boundary layer (ABL) growth after the inversion was overcome in these perturbation cases. The ABL growth could then support the mixing of the trapped moisture and potentially decrease the LCL to the PBLH.

Given a wet soil, we would expect high surface evaporation moistening the boundary layer below the inversion in the morning hours, but also weak sensible heat fluxes and thus a weak "force" pushing against the inversion. Hence, weak BL growth would be expected in the morning hours over wet soils. This would hint to a longer timescale to overcome the inversion, which – following Ek and Mahrt (1994) – would lead to less clouds. Over dry soils, we would expect higher sensible heat fluxes and thus a higher probability to overcome the inversion and foster BL growth. Although, there is less moistening of the boundary layer from the surface expectable, it is more likely that the inversion is overcome and thus a coupling event could occur.

So, the case you referred to may indeed occur, but a dry coupling event seems more likely. However, whether a coupling event could occur over wet soil is expected to be strongly dependent on the inversion strength and whether there is still sufficient boundary layer heating to overcome the inversion early enough to enable the development of clouds. Also the entrainment flux plays a role for the boundary layer development and the L-A coupling signal (van Heerwaarden et al., 2009). This information is not included in the framework without further extensions, as it neither considers the lowest 1000m of the BL, nor entrainment, nor the energy partitioning at the land surface. Hence, the CTP-HI$_{low}$ framework would not capture the effects of changes in inversion strength in any case.

van Heerwaarden, C. C., Vilà-Guerau de Arellano, J., Moene, A. F., and Holtslag, A. A. M.: Interactions between dry-air entrainment, surface evaporation and convective

boundary-layer development: DRY-AIR ENTRAINMENT, SURFACE EVAPORATION AND CBL DEVELOPMENT, Q.J.R. Meteorol. Soc., 135, 1277–1291, https://doi.org/10.1002/qj.431, 2009.

L355-357: Since you are discussing the influence of temperature in the "hot case," it can be a bit confusing when you then say "fraction of nAC-days within the hotspot" as my mind first thought hotspot in a literal, temperature sense. I would suggest changing this to read: "within the L-A coupling hotspot."

Thank you for bringing up this point. We agree that the term hotspot can be misinterpreted in association with the hot case. We have revised the text according to your suggestion:

L393-395: *"Hot case: Causes a higher temperature and temperature gradient between 100-300 hPa AGL with corresponding changes in moisture. These lead to greater instability with a constant humidity deficit, which increases the expansion of the hotspot and the fraction of nAC days within the L-A coupling hotspot."*

L360-362: Is this where CTP is increased, but so is the temperature inversion? So perhaps the likelihood of convective triggering over wet soils could be tied to the comments associated with L343-346 above?

In the north, the increase in CTP - implying a destabilization of the atmosphere above the inversion - rather pushes stable (and eventually wet) atmospherically controlled days to the non-atmospherically controlled regime.

Following our argumentation in the response to your comment above, the framework likely does not represent changes in the inversion strength, because the integration of CTP starts at 100hPa AGL. Hence, the process mentioned above would not be captured by the framework. Nevertheless, it still may occur, but we think that further analyses are necessary, which e.g. involve CIN to represent the inversion strength and additionally the surface fluxes as an indicator of whether the inversion can be overcome before the end of the mixed-layer development.

Broader comment regarding discussion: How might overall warming of the climate impact the length of the season in which we consider L-A coupling to be most influential? Your results imply that warming enhances coupling strength, so would that also mean that L-A coupling might be and important driver of hydroclimatic variability over a longer warm season? For example, instead of JJA, perhaps the "coupling season" would now be MJJAS?

Dirmeyer et al. (2013) suggested an earlier springtime onset of L-A feedbacks over the US in the future. So given that warming enhances the coupling strength, a longer warm

season might indeed also imply a prolonging of the "coupling season" to MJJAS over Europe. However, our analyses were not tailored to investigate this effect and do not provide enough evidence to give a sound answer to your questions.

Dirmeyer, P. A., Jin, Y., Singh, B., and Yan, X.: Trends in Land–Atmosphere Interactions from CMIP5 Simulations, 14, 829–849, https://doi.org/10.1175/JHM-D-12-0107.1, 2013.

L466-468: I wholeheartedly agree that we need more vertical resolution, everywhere, however, I do believe you can argue that while Wakefield et al. (2021) shows that vertical resolution is a limiting factor, you can still get representative estimates of the L-A coupling pre-conditioning even when vertical resolution is unfortunately limited. Therefore, I think you can use this reference to argue both points… your limitation in vertical resolution introduces uncertainty, but that uncertainty is not so large that it substantially impacts the validity of your results.

Thank you for bringing up this line. We will gladly pick it up in the discussion section:

L519-521: *"However, while they on the one hand show that lower vertical resolution introduces uncertainty, they also showed that data with limited resolution still provide reasonable results. Thus, the effects of altered gradients are expected to remain substantially similar also with a higher vertical resolution model output."*

L479: I'm not sure about the use of the word "reliable." My mind immediately jumps to an operational use of the word and thinking about model reliability. I do like that you say the feedback class is *insensitive* to changes though. Maybe "wherefore a consistent regime can be expected," if that's the message you are trying to convey.

Thank you for this suggestion. We have revised the sentence according to it:

L540-541: *"There are two regions in which the dominant feedback class is insensitive to changes in the atmospheric structure, wherefore a consistent regime can be expected."*

Technical:

L123: Change "but maybe limit investigations" to "may limit investigations."

L142: Change "deep convection is inhibited by an inversion, only shallow clouds…" to "deep convection is inhibited by an inversion and only shallow clouds…" or separate into two sentences.

L382: Typo "Please not that…" should say. "Please note that…"

We have incorporated all of your technical suggestions.